# Cold-responsive transcription factors in Arabidopsis and rice: A regulatory network analysis using array data and gene co-expression network

**Khazar Edrisi Maryan**[1,2]*, **Naser Farrokhi**[1]*, **Habibollah Samizadeh Lahiji**[2]

**1** Department of Cell & Molecular Biology, Faculty of Life Sciences & Biotechnology, Shahid Beheshti University, Tehran, Iran, **2** Department of Plant Biotechnology, Faculty of Agriculture, University of Guilan, Rasht, Iran

* n_farrokhi@sbu.ac.ir (NF); khazaredrisi@gmail.com (KEM)

## Abstract

Plant growth and development can be influenced by cold stress. Responses of plants to cold are regulated in part by transcription factors (TFs) and microRNAs, which their determination would be necessary in comprehension of the corresponding molecular cues. Here, transcriptomes of Arabidopsis and rice were analyzed to computationally determine TFs and microRNAs that are differentially responsive to cold treatment, and their co-expression networks were established. Among 181 Arabidopsis and 168 rice differentially expressed TF genes, 37 (26 novel) were up- and 16 (8 novel) were downregulated. Common TF encoding genes were from ERF, MYB, bHLH, NFY, bZIP, GATA, HSF and WRKY families. NFY A4/C2/A10 were the significant hub TFs in both plants. Phytohormone responsive *cis*-elements such as ABRE, TGA, TCA and LTR were the common *cis*-elements in TF promoters. Arabidopsis had more responsive TFs compared to rice possibly due to its greater adaptation to ranges geographical latitudes. Rice had more relevant miRNAs probably because of its bigger genome size. The interacting partners and co-expressed genes were different for the common TFs so that of the downstream regulatory networks and the corresponding metabolic pathways. Identified cold-responsive TFs in (A + R) seemed to be more engaged in energy metabolism *esp.* photosynthesis, and signal transduction, respectively. At post-transcriptional level, miR5075 showed to target many identified TFs in rice. In comparison, the predictions showed that identified TFs are being targeted by diverse groups of miRNAs in Arabidopsis. Novel TFs, miRNAs and co-expressed genes were introduced as cold-responsive markers that can be harnessed in future studies and development of crop tolerant varieties.

## Introduction

Understanding how plants respond to cold stress will provide valuable information and lay grounds to identify genetic resources in crop tolerance improvement [1]. A set of transcription

**Data Availability Statement:** The datasets analyzed during the current study are available in the Gene Expression Omnibus (GEO) repository, with the accession numbers; "GSE33978",

"GSE63184", "GSE5536", "GSE3326", "GSE86605", "GSE63131", "GSE41935", "GSE38030", "GSE38023", "GSE71680", "GSE83912", "GSE37940", "GSE32065", "GSE19983", "GSE32704", and "GSE6901" and related article of each accession number are listed in "References", from reference number 89 to reference number 104, respectively.

**Funding:** The author(s) received no specific funding for this work.

**Competing interests:** The authors have declared that no competing interests exist.

**Abbreviations:** TF, Transcription factor.

factors (TFs) either working together or independently can regulate the downstream cold-responsive genes to adjust plant cells and organs by means of biochemical and physiological alterations [2]. TFs establish the whole functional networks [3], and more specifically modulate gene expression in response to abiotic stresses [4]. Identification of cold-induced TFs helps in developing better management systems on the genes that directly (*e.g.*, anti-freeze proteins and osmo-regulators) or indirectly (*e.g.*, chaperones and kinases) act in the favor of cold tolerance [5]. Prediction of TFs' function by means of bioinformatics has become an important and effective strategy to reveal details of complicated biological systems [6].

In the past two decades, the molecular mechanisms of cold stress responses have been extensively studied in (A + R). A well-known transcriptional regulatory pathway involved in plant cold adaptation is the CBF/DREB1 cold signalling pathway, mediated by CBF TFs [7]. Cold-activated CBF TFs (CBF1 to CBF3) induce cold response genes; recognizing and binding to the C-repeat/dehydration responsive element (CRT/DRE) motif in the promoters of many cold-responsive (COR) genes, such as *RD29a* and *COR15a*. In order to promote the transcription of the downstream genes [8]. These genes are strongly upregulated in a CBF-dependent manner to enhance the freezing resistance, by stabilizing the chloroplast membranes when constitutively (over) expressed [9]. Vyse et al. [10] reported four TFs, CBF2/DREB1C, CBF4/DREB1D, DDF2/DREB1E and DDF1/DREB1F to be uniquely and significantly induced throughout the entire cold response. Given the fact that only ~12% of the cold-regulated genes are being regulated by CBFs [11], one has to assume that other TFs are of importance for plant cold acclimation.

Direct evidence exists for the activities of some prominent cold-regulated TF families not participating in the CBF cold response pathway, such as basic helix-loop-helix (bHLH) [10, 12], MYB [13], basic region/leucine zipper motif (bZIP) [14], WRKY33 [15], and NAC [16]. Park et al. [17] identified 30 TFs in Arabidopsis that are quickly induced by low temperatures, and these genes were named "first-wave cold-inducible TFs", namely ZAT12, HSFC1, RAV1, MYB73, MYB44, CRF2, WRKY33, ERF6, CRF3 and RVE2. In rice, corresponding TFs were OsIAA23, SNAC2, OsWRKY1v2, OsWRKY1v24, OsWRKY1v53, OsWRKY1v71, HMGB, OsbHLH, and OsMyb [18]. Zeng et al. [19], used RNA-seq technology to analyze the cold stress and recovery process in rice, which identified different TFs in different modules including different numbers of bZIP, HOX, AP2-EREBP, MADS, MYB, NAC, TIFY and WRKY.

Here, Arabidopsis and rice (A + R) *in silico* transcriptome analysis were performed under cold stress conditions in order to pinpoint the regulatory elements involved. The aim was to provide a comprehensive overview of common/differential molecular mechanisms and gene regulatory networks in response to cold stress (S1 Graphical abstract). There were 286 and 403 GO Datasets (Array and SRA) in NCBI database (https://www.ncbi.nlm.nih.gov/gds/term=cold stress, 2021) with different cold and freezing temperature treatments in different growth stages and tissues of (A + R), respectively. Amongst which, sixteen GO datasets with the exact same cold treatment (0–5°C) and tissue (seedling) in both plants were chosen to be analyzed.

In addition to TFs, microRNAs (miRNAs) next to other means such as circular RNAs, long non-coding RNAs, and microproteins are the other regulatory factors [20]. Different findings indicate the common roles of many miRNAs regulatory elements to be responsive in multiple stress conditions [21]. Some reports for miRNA and their predicted targets involved in the regulation of growth and development under low-temperature stress have already been presented. Overexpression of miR1320 resulted in increased cold tolerance in rice. AP2/ERF TF OsERF096, as a target of miR1320, co-regulate cold tolerance by repressing the JA-mediated cold signalling pathway [22]. Similarly, overexpression of Osa-miR156, Osa-miR319, and Osa-miR528 improved cold tolerance in rice [23]. In addition, miR319 positively regulates cold tolerance by targeting *OsPCF6* and *OsTCP21* transcripts in rice, and the downregulation of these

two TFs resulted in enhanced tolerance to cold stress [24]. Zhang et al. 2022 [25], reported that Aux/IAA14 regulates miRNA-mediated cold stress responding mechanism in Arabidopsis roots. Based on next-generation sequencing, 180 known and 71 novel cold-responsive miR-NAs were revealed. Comparative analysis of miRNA expression showed notable difference of 13 known and 7 novel miRNAs in *slr1* (mutation in Aux/IAA14) and WT. Interestingly, compared with WT, miR169 was downregulated in *slr1* after 12-h of cold treatment at 4°C, of particularly interest was miR169a, miR169d, and miR169h.

Stress-response miRNA studies can provide important understanding in plant stress resistance breeding and gene expression, a powerful approach to unravel new insight into adaptive mechanism in plants [25]. Therefore, here the interaction between miRNAs and TFs was studied. Moreover, to further corroborate on the function of *trans*-elements, in our case TFs and miRNAs, co-expression analyses of the candidate TFs were carried out [26]. Due to the diversity and a large number of co-expressed genes, protein kinase genes were chosen for further promoter analysis to figure out if identified TFs could induce their transcription in cold stress.

## Methods

### Differentially expressed TFs

The gene expression data of eight cold-treated microarray datasets were retrieved from GEO [27] for *A. thaliana* and *O. sativa* in seedling stage treated for 24 h at 4–5°C (S1 Table). GEO2R was used to profile individual dataset lists of transcripts with significant increase and decrease in abundance compared to the untreated control condition. Differentially expressed transcripts (DETs) of TFs were defined with greater than two-fold change compared to the controls. LogFC signal intensities with false discovery rate were adjusted by *p*-value < 0.05 (Weltch t-test with Benjamini-Hochberg correction). Common up- and down-regulated TFs between the two model plants were identified using Venny 2.1.0 [28] to draw a Venn diagram (the S1 Graphical abstract, Table 1) and uncommon TFs are listed in S2 Table.

Protein sequences of candidate TFs were downloaded from Uniprot [29] in FASTA format. Analysis of conserved domains was performed using MEME [30] and SALAD [31] (Fig 8A and 8B). Due to better motif representation and production of dendrogram, the results of SALAD are presented.TF properties were uploaded from the Plant Transcription Factor Database [32] (S3 Table). Homology searches for TFs were carried out using BLASTP [33] (S4 Table). TF regulatory interactions were retrieved from CORNET [34] using both experimental and predicted data of IntACt, TAIR and AtPID regulatory interactions (S5 Table).

### miRNAs target TF transcripts

The coding sequences of (A + R) TFs were downloaded from Gene at NCBI [35] and Rice Genome Annotation Project database [36], respectively. miRNAs that target the corresponding transcripts were fetched from psRNATarget [37]. Identified miRNAs were confirmed through checking the thermodynamic stability of mRNA-miRNA hybrid based on the minimum free energy (Mfe) using RNAhybrid [38] (Table 2 and S6 Table). The mechanism of action was determined according to Liu et al. (2014) [39], mRNA hydrolysis with less than four nucleotide mismatches, and halt in translation with greater than four nucleotide mismatches between miRNA and the target sequence and in our case the candidate TF transcript.

### Gene expression profile

Hierarchical cluster analysis was carried out to illustrate TF gene expression via heatmap using R statistical language [40] at pheatmap package [41] (Fig 1). Co-expressed genes for each TF

**Table 1. Identified common up- and down- regulated TFs in (A + R) in response to cold.**

| UP-regulated | | | | Down-regulated | | | | |
|---|---|---|---|---|---|---|---|---|
| TF Family | No. | TF Name | Arabidopsis accession No. | Rice LOCUS ID | TF Family | No. | TF Name | Arabidopsis accession No. | Rice LOCUS ID |
| AP2-like ethylene-responsive | 1 | ANT | AT4G37750 | Os03g12950 | AP2-like ethylene-responsive | 1 | PLT2 | At1g51190 | Os06g44750 |
| Ethylene-responsive (ERF) | 2 | ERF 4 | AT3G15210 | Os12g39330 | Ethylene-responsive (ERF) | 2 | ERF39 | AT4G16750 | Os01g10370 |
| | 3 | ERF 5 | AT5G47230 | Os07g10410 | | 3 | ERF54 | AT4G28140 | Os01g46870 |
| | 4 | ERF13 | AT2G44840 | Os06g11940 | MYB | 4 | MYB5 | AT3G13540 | Os05g41166 |
| | 5 | ERF38 | AT2G35700 | Os02g13710 | | 5 | MYB37/RAX1 | AT5G23000 | Os01g09590 |
| | 6 | ERF73 | AT1G72360 | Os09g11460 | | 6 | MYB38/RAX2 | AT2G36890 | Os02g42870 |
| | 7 | ERF74-RAP2-12 | AT1G53910 | Os05g41780 | | 7 | MYB44 | AT5G67300 | Os01g52410 |
| | 8 | ERF98 | AT3G23230 | Os02g34260 | | 8 | MYB84/RAX3 | AT3G49690 | Os09g01960 |
| | 9 | ERF113 | AT5G13330 | Os06g42990 | bHLH | 9 | bHLH112 | AT1G61660 | Os04g53990 |
| | 10 | DREB 1A | AT4G25480 | Os09g35030 | | 10 | bHLH113 | AT3G19500 | Os03g55220 |
| | 11 | DREB 1B/CBF1 | AT4G25490 | Os09g35010 | Nuclear transcription factor Y subunit | 11 | NF-Y B-3 | AT4G14540 | Os05g49780 |
| MYB | 12 | MYB57 | AT3G01530 | Os02g40530 | | 12 | NF-Y B-4 | AT1G09030 | Os05g38820 |
| | 13 | MYB59 | AT5G59780 | Os01g74410 | | 13 | NF-Y B-9 | AT1G21970 | Os06g17480 |
| bHLH | 14 | bHLH16/UNE10 | AT4G00050 | Os07g36460 | | 14 | NF-Y C-2 | AT1G56170 | Os03g14669 |
| | 15 | bHLH35 | AT5G57150 | Os04g23550 | bZIP | 15 | bZIP17 | AT2G40950 | Os02g10140 |
| | 16 | bHLH 59/UNE12 | AT4G02590 | Os02g02480 | TCP1 | 16 | TCP21 | AT5G08330 | Os07g05720 |
| | 17 | bHLH79 | AT5G62610 | Os02g47660 | | | | | |
| | 18 | bHLH102/BIM2 | AT1G69010 | Os12g41650 | | | | | |
| | 19 | bHLH105/ILR3 | AT5G54680 | Os08g04390 | | | | | |
| | 20 | bHLH116/ICE1 | AT3G26744 | Os01g50940 | | | | | |
| | 21 | bHLH128 | AT1G05805 | Os07g39940 | | | | | |
| | 22 | bHLH129 | AT2G43140 | Os03g10770 | | | | | |
| | 23 | bHLH137 | AT5G50915 | Os08g42470 | | | | | |
| | 24 | bHLH148 | AT3G06590 | Os03g53020 | | | | | |
| Nuclear transcription factor Y subunit | 25 | NFYA-4 | AT2G34720 | Os03g48970 | | | | | |
| | 26 | NFYA-10 | AT5G06510 | Os12g42400 | | | | | |
| bZIP | 27 | bZIP20/TGA2 | AT5G06950 | Os01g59350 | | | | | |
| | 28 | bZIP45/TGA6 | AT3G12250 | Os05g49420 | | | | | |
| | 29 | bZIP 60 | AT1G42990 | Os07g44950 | | | | | |
| GATA | 30 | GATA 11 | AT1G08010 | Os02g12790 | | | | | |
| | 31 | GATA 22 | AT4G26150 | Os06g37450 | | | | | |
| | 32 | GATA 23 | AT5G26930 | Os01g24070 | | | | | |
| Heat shock | 33 | HSF A-3 | AT5G03720 | Os02g32590 | | | | | |
| | 34 | HSF A-9 | AT5G54070 | Os03g12370 | | | | | |
| | 35 | HSF B-2b | AT4G11660 | Os08g43334 | | | | | |
| | 36 | HSF B4 | AT1G46264 | Os07g44690 | | | | | |
| WRKY | 37 | WRKY1/ZAP1 | AT2G04880 | Os01g14440 | | | | | |

**Table 2. Predicted miRNAs targeting TFs in (A + R).** The prediction was done using psRNATarget [37] and RNAhybrid [38].

| TF name | *ath-miRNA* | Mfe (kcal/mol) | *osa* -miRNA | Mfe (kcal/mol) |
|---|---|---|---|---|
| ANT | *ath-miR5020c* | -27.0 | *osa-miR6255* | -28.9 |
| ERF 4 | *ath-miR5646* | -28.2 | *osa-miR531a,c* | -43.1 |
| | | | *osa-miR2926* | -32.2 |
| | | | *osa-miR2927* | 39.2 |
| | | | *osa-miR5075* | -42.6 |
| | | | *osa-miR1437b-3p* | -42.4 |
| | | | *osa-miR1848* | -41.9 |
| | | | *osa-miR531b* | -36.7 |
| ERF 5 | *ath-miR414* | -28.8 | osa-miR437 | -22.3 |
| ERF13 | *ath-miR391-5p* | -28.2 | *osa-miR2094-3p* | -38.1 |
| | | | *osa-miR5075* | -37.3 |
| | | | *osa-miR5485* | -33.1 |
| | | | *osa-miR156c-3p* | -32.4 |
| | | | *osa-miR156f, h, I-3p* | -30.7 |
| ERF38 | *ath-miR414* | -28.1 | *osa-miR439a to i* | -37.2 |
| | | | *osa-miR5792* | -34.2 |
| | | | *osa-miR1846a,b,c,-5p* | -36.4 |
| | | | *osa-miR1846d-3p* | -34.2 |
| | | | *osa-miR2927* | -33.5 |
| | | | *osa-miR319a-3p* | -35.7 |
| ERF73 | *ath-miR8168* | -34.7 | osa-miR5075 | -46.6 |
| | | | osa-miR156c,g,-3p | -36.2 |
| | | | osa-miR530-3p | -34.3 |
| | | | osa-miR156f,h,I-3p | -33.7 |
| | | | osa-miR2096-5p | -38.5 |
| | | | osa-miR2919 | -30.4 |
| | | | osa-miR528-3p | -33.0 |
| ERF74-RAP2-12 | *ath-miR396a-5p,b* | -29.1 | osa-miR2102-3p | -42.7 |
| | | | osa-miR2924 | -36.3 |
| | | | osa-miR5496 | -33.9 |
| | | | osa-miR3980a,b-3p | -34.9 |
| ERF98 | *ath-miR3932a,b* | -29.2 | osa-miR5075 | -39.9 |
| | | | osa-miR5540 | -32.2 |
| ERF113 | *ath-miR5021* | -25.4 | osa-miR5075 | -43.2 |
| | | | osa-miR3979-5p | -39.7 |
| | | | osa-miR6249a,b | -34.3 |
| | | | osa-miR5530 | -33.8 |
| | | | osasa-miR396a,b-5p | -30.2 |
| | | | osa-miR396c-5p | -30.4 |
| | | | osa-miR408-3p | -34.8 |
| DREB 1A | *ath-miR5020c* | - | osa-miR5075 | -46.2 |
| | | | osa-miR2927 | -34.9 |
| | | | osa-miR5489 | -33.4 |
| | | | osa-miR5495 | -31.0 |
| | | | osa-miR5514 | -33.6 |

*(Continued)*

**Table 2.** (Continued)

| TF name | *ath-miRNA* | Mfe (kcal/mol) | *osa*-miRNA | Mfe (kcal/mol) |
|---|---|---|---|---|
| DREB 1B | *ath-miR5646* | - | *osa-miR5075* | -41.2 |
| | | | *osa-miR528-3p* | -38.4 |
| | | | *osa-miR5795* | -34.4 |
| | | | *osa-miR5819* | -39.6 |
| MYB57 | *ath-miR5654-3p* | -31.9 | *osa-miR5832* | -35.4 |
| | *ath-miR858a* | -31.8 | *osa-miR529a* | -34.3 |
| | *ath-miR858b* | -30.7 | *osa-miR5075* | -35.0 |
| MYB59 | *ath-miR858a,b* | -28.0 | *osa-miR5833* | -39.2 |
| | | | *osa-miR1858a,b* | -35.9 |
| | | | *osa-miR2926* | -34.8 |
| bHLH16/ UNE10b | *ath-miR838* | -23.2 | *osa-miR5075* | -46.5 |
| | | | *osa-miR1846a,b-3p* | -34.2 |
| | | | *osa-miR159a.2* | -31.0 |
| bHLH35 | *ath-miR1886.1* | -26.5 | *osa-miR414* | -29.3 |
| bHLH 59/ UNE12 | *ath-miR472-3p* | -30.2 | *osa-miR5075* | -36.9 |
| bHLH79 | *ath-miR779.2* | -21.9 | *osa-miR5075* | -40.7 |
| | | | *osa-miR2926* | -31.3 |
| | | | *osa-miR319a-3p* | -31.4 |
| | | | *osa-miR5832* | -36.3 |
| bHLH102/BIM2 | *ath-miR867* | -21.2 | *osa-miR5075* | -46.9 |
| | | | *osa-miR1846a,b-3p* | -36.3 |
| | | | *osa-miR1879* | -33.7 |
| | | | *osa-miR2926* | -35.1 |
| bHLH105/ ILR3 | *ath-miR3440b-3p* | -30.1 | *osa-miR1846a,b-3p* | -36.3 |
| | | | *osa-miR1879* | -33.7 |
| BHLH116/ICE1 | *ath-miR156a-3p* | -29.1 | *osa-miR2926* | -35.1 |
| bHLH128 | *ath-miR395a,b,c,d,e,f* | -30.5 | *osa-miR408-3p* | -32.5 |
| bHLH129 | *ath-miR779.1* | -33.0 | *osa-miR5493* | -39.8 |
| | | | *osa-miR5075* | -39.0 |
| | | | *osa-miR5809* | -37.8 |
| | | | *osa-miR5832* | -36.8 |
| | | | *osa-miR6249a,b* | -36.6 |
| | | | *osa-miR2927* | -33.2 |
| | | | *osa-miR2097-3p* | -31.7 |
| bHLH137 | *ath-miR5023* | -29.9 | *osa-miR5515* | -33.1 |
| bHLH148 | *ath-miR870-3p* | -25.5 | *osa-miR5075* | -45.4 |
| | | | *osa-miR529a* | -33.3 |
| NFYA-4 | - | - | *osa-miR5075* | -38.4 |
| | | | *osa-miR1862a,b,c* | -33.7 |
| NFYA-10 | *ath-miR836* | -27.8 | *osa-miR2873a* | -21.1 |
| bZIP20/TGA2 | *ath-miR3434-3p* | -27.3 | *osa-miR5075* | -42.0 |
| | | | *osa-miR1437b-3p* | -39.1 |
| | | | *osa-miR2094-5p* | -34.9 |
| bZIP45/TGA6 | *ath-miR859* | -26.2 | *osa-miR171i-5p* | -28.4 |
| | | | *osa-miR172d-5p* | -28.4 |
| bZIP 60 | *ath-miR414* | -29.3 | *osa-miR5795* | -35.1 |
| | | | *osa-miR5527* | -31.9 |
| | | | *osa-miR5832* | -32.3 |

(*Continued*)

**Table 2.** (Continued)

| TF name | *ath-miRNA* | Mfe (kcal/mol) | *osa*-miRNA | Mfe (kcal/mol) |
|---|---|---|---|---|
| GATA 11 | *ath-miR5020a* | -25.2 | *osa-miR2925* | -34.0 |
| GATA 22 | *ath-miR8172* | -24.3 | *osa-miR528-3p* | -37.7 |
| | | | *osa-miR5833* | -34.8 |
| | | | *osa-miR5809* | -34.4 |
| | | | *osa-miR5075* | -33.8 |
| | | | *osa-miR1865-5p* | -32.1 |
| | | | *osa-miR2927* | -32.6 |
| | | | *osa-miR439a,b,c,d,e,f,g,h,i* | -32.3 |
| GATA 23 | *ath-miR5020b* | -24.3 | *osa-miR168b* | -32.3 |
| HSF A-3 | *ath-miR156b-3p* | -30.9 | *osa-miR5075* | -34.1 |
| | | | *osa-miR1858a,b* | -32.2 |
| | | | *osa-miR2927* | -31.7 |
| HSF A-9 | *ath-miR395b,c,f* | -30.1 | *osa-miR5075* | -42.4 |
| | | | *osa-miR156b-3p* | -34.5 |
| HSF B-2b | *ath-miR834* | -37.3 | *osa-miR5075* | -42.4 |
| | *ath-miR395b,c,f* | -31.1 | *osa-miR156b-3p* | -34.5 |
| HSF B4 | - | - | *osa-miR5075* | -41.6 |
| | | | *osa-miR6249a,b* | -34.3 |
| | | | *osa-miR1847.1* | -31.5 |
| | | | *osa-miR160a,b,c,d-5p* | -34.0 |
| | | | *osa-miR160e-5p* | -34.6 |
| WRKY1/ZAP1 | *ath-miR771* | -31.6 | *osa-miR5493* | -42.7 |
| | | | *osa-miR531a,c* | -43.1 |
| | | | *osa-miR531b* | -38.6 |
| | | | *osa-miR2927* | -37.0 |
| | | | *osa-miR5075* | -40.5 |
| | | | *osa-miR1848* | -40.0 |
| | | | *osa-miR2925* | -35.3 |
| PLT2 | *ath-miR5658* | -28.2 | *osa-miR2094-5p* | -33.6 |
| | | | *osa-miR2864.1* | -30.8 |
| | | | *osa-miR164d* | -30.3 |
| ERF39 | *ath-miR855* | -24.0 | *osa-miR5075* | -41.8 |
| | | | *osa-miR530-3p* | -34.7 |
| | | | *osa-miR1858a,b* | -34.3 |
| ERF54 | *ath-miR5020a* | -26.3 | *osa-miR6246* | -37.5 |
| | | | *osa-miR5150-3p* | -34.6 |
| | | | *osa-miR530-3p* | -34.3 |
| | | | *osa-miR5493* | -36.7 |
| MYB5 | *ath-miR171c-5p* | -24.5 | *osa-miR159c* | -41.9 |
| | | | *osa-miR159d* | -41.2 |
| | | | *osa-miR159e* | -40.7 |
| | | | *osa-miR159a.1* | -38.8 |
| | | | *osa-miR159b* | -38.8 |
| | | | *osa-miR159f* | -39.1 |
| | | | *osa-miR319a-3p.2-3p* | -31.1 |
| | | | *osa-miR319b* | -31.1 |

(*Continued*)

**Table 2.** (Continued)

| TF name | *ath-miRNA* | Mfe (kcal/mol) | *osa*-miRNA | Mfe (kcal/mol) |
|---|---|---|---|---|
| MYB37/RAX1 | | | *osa-miR2104* | -46.3 |
| | | | *osa-miR2925* | -41.3 |
| | | | *osa-miR5075* | -41.1 |
| | *ath-miR858a,b* | -32.8 | *osa-miR5809* | -36.8 |
| | *ath-miR8167a,b,c,d,e,f,* | -30.4 | *osa-miR6249a,b* | -33.7 |
| | | | *osa-miR1865-5p* | -31.9 |
| | | | *osa-miR2926* | -31.1 |
| | | | *osa-miR2927* | -32.8 |
| MYB38/RAX2 | *ath-miR5658* | -31.3 | *osa-miR2925* | -41.2 |
| | | | *osa-miR3979-3p* | -33.3 |
| | | | *osa-miR166k-5p* | -33.3 |
| | | | *osa-miR1848* | -37.4 |
| | | | *osa-miR2102-5p* | -40.3 |
| | | | *osa-miR2919* | -30.7 |
| | | | *osa-miR2926* | -31.8 |
| MYB44 | *ath-miR5016* | -31.4 | *osa-miR2927* | -34.1 |
| | | | *osa-miR5078* | -32.9 |
| | | | *osa-miR5809* | -32.8 |
| MYB84/RAX3 | *ath-miR858a,b* | -32.9 | *osa-miR1846a-5p,b,c* | -37.6 |
| | *ath-miR414* | -32.8 | *osa-miR1851* | -31.5 |
| bHLH112 | *ath-miR391-5p* | -29.9 | *osa-miR531a,c* | -44.9 |
| | | | *osa-miR531b* | -42.4 |
| | | | *osa-miR2102-5p* | -43.1 |
| | | | *osa-miR5832* | -32.8 |
| | | | *osa-miR1865-5p* | -31.3 |
| | | | *osa-miR2926* | -33.8 |
| | | | *osa-miR2927* | -35.4 |
| | | | *osa-miR5075* | -40.8 |
| bHLH113 | *ath-miR771* | -33.9 | *osa-miR5075* | -37.2 |
| | | | *osa-miR5809* | -33.7 |
| | | | *osa-miR5150-3p* | -34.7 |
| NF-Y B-3 | *ath-miR157c-3* | -26.8 | *osa-miR5075* | -45.7 |
| NF-Y B-4 | *ath-miR5012* | -23.8 | *osa-miR5833* | -42.9 |
| | | | *osa-miR2926* | -30.3 |
| | | | *osa-miR156j-3p* | -33.7 |
| NF-Y B-9 | *miR854a,b,c,d,e* | -33.9 | *osa-miR531a,c* | -45.5 |
| | | | *osa-miR5832* | -37.9 |
| | | | *osa-miR531b* | -38.1 |
| NF-Y C-2 | *ath-miR414* | -28.5 | *osa-miR5484* | -37.0 |
| | | | *osa-miR156j-3p* | -33.8 |
| | | | *osa-miR2864.1* | -31.9 |
| | | | *osa-miR6249a,b* | -36.0 |
| bZIP17 | *ath-miR834* | -29.5 | *osa-miR5075* | -45.7 |
| | | | *osa-miR2927* | -35.0 |

(*Continued*)

**Table 2.** (Continued)

| TF name | *ath-miRNA* | Mfe (kcal/mol) | *osa*-miRNA | Mfe (kcal/mol) |
|---------|-------------|----------------|-------------|----------------|
| TCP21 | *ath-miR5658* | -26.4 | *osa-miR2925* | -41.9 |
| | | | *osa-miR319a-3p.2-3p* | -38.9 |
| | | | *osa-miR319b* | -38.9 |
| | | | *osa-miR159c,d* | -36.5 |
| | | | *osa-miR159e* | -36.9 |
| | | | *osa-miR159a.1* | -34.0 |
| | | | *osa-miR159b* | -34.0 |
| | | | *osa-miR159f* | -33.9 |
| | | | *osa-miR2927* | -38.7 |
| | | | *osa-miR1846a,b,c-5p* | -37.3 |
| | | | *osa-miR2926* | -33.5 |
| | | | *osa-miR5150-5p* | -34.4 |

were retrieved from AttedII [42] for Arabidopsis and RiceFREND [43] for rice according to MR values greater than 50 (Tables will be provided upon request). RiceDB [44] was used to obtain locus link identifiers. Phytohormonal control of TFs was checked at PlantTFDB [32] (S7 Table). Gene Ontology enrichment analysis of co-expressed genes was carried out to determine the molecular function, biological process, and cellular component using PANTHER [45] (Fig 6, S8 Table).

Promoter sequences of TFs (S9 Table) and their co-expressed protein kinase genes were retrieved from Plant promoter database [46] and promoter analysis of TFs (S9 Table) and their co-expressed protein kinase genes (S10 Table) was performed using 1000 bp upstream sequences through PLANTCARE [47], PlantPAN [48] and AGRIS AtcisDB [49]. Metabolic pathway analysis of co-expressed genes was carried out at KEGG pathway database [50] (S11–S14 Tables).

### Protein- protein interactions

Protein-protein interaction networks (PPI) of TFs were created using STRING [51] (Figs 2–5, 7, S13 and S14 Tables). Cytoscape software [52] was used for visualizing the interaction networks (Figs 2–5 and 7). In a co-expression network, Maximal Clique Centrality (MCC) algorithm was reported to be the most effective method of finding hubs [53]. The MCC of each node was calculated by CytoHubba, a plugin in Cytoscape [54] (Figs 2–5 and 7). In this study, the genes with the top 10 MCC values were considered as the hub genes.

## Results

### Differentially expressed TFs

TF expression patterns were checked in 16 separate microarrays datasets, eight for Arabidopsis and eight for rice during cold stress (S1 Table; Fig 1). TFs with increased in abundance in response to cold were 119 for Arabidopsis and 86 for rice. Common TF families (including 37 TFs between the two model plants) belonged to AP2, ERF, MYB, bHLH, NF-Y, GATA, HSF, and WRKY (Table 1). TFs with decreased in abundance were 62 in Arabidopsis and 82 in rice. The common downregulated TFs (16 genes) belonged to AP2, ERF, MYB, bHLH, NF-Y, bZIP, TCP, and trihelix TF families (Table 1). Here, we report 26 novel upregulated TFs including ANT, ERF74, and ERF98 (ERF family), MYB57 and MYB59 (MYB family), bHLH16, bHLH59, bHLH79, bHLH102, bHLH105, bHLH128, bHLH129, bHLH137, and bHLH148

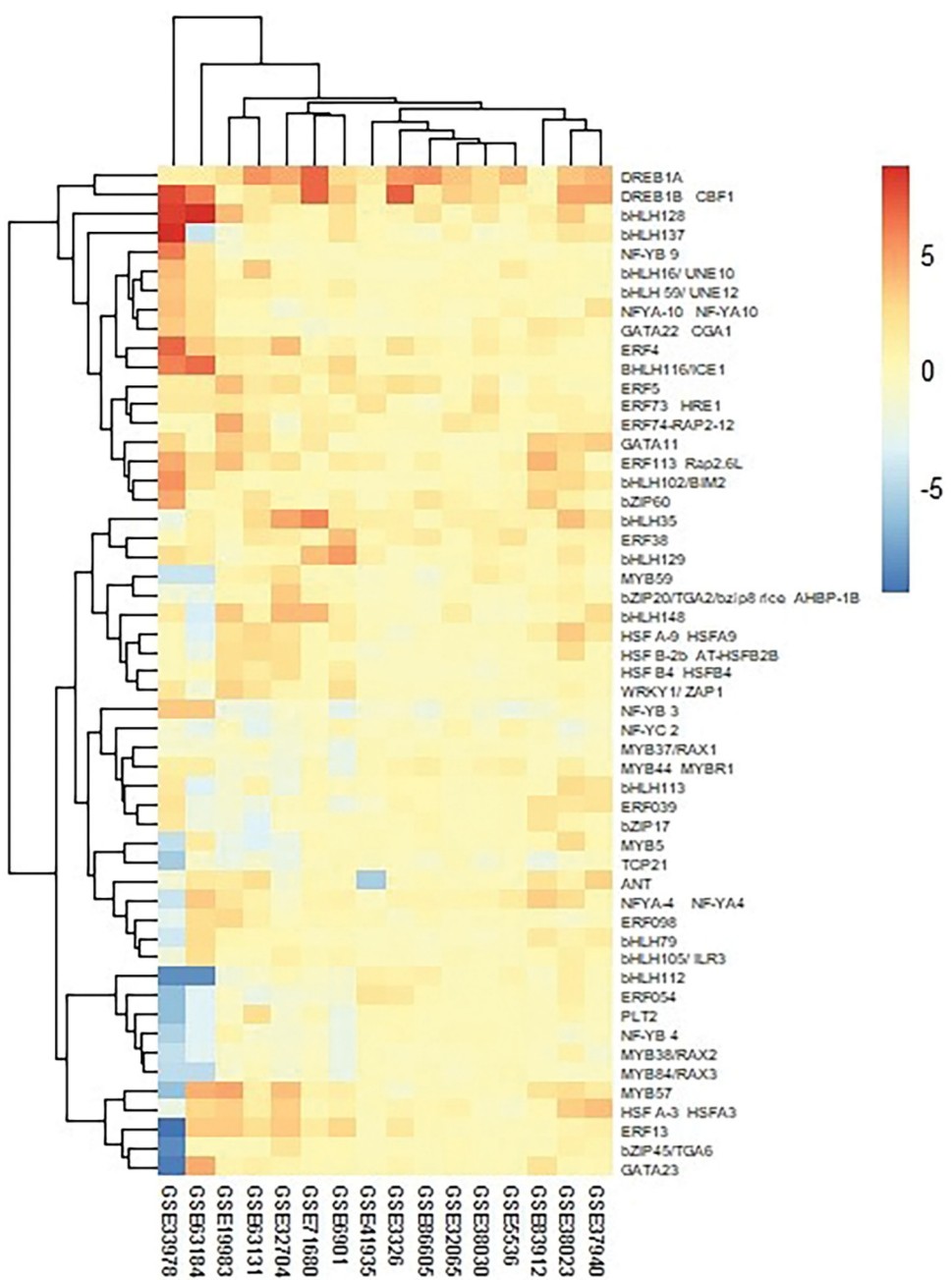

**Fig 1. Hierarchical cluster analysis was carried out to illustrate the gene expression profile of up-and down-regulated TFs in (A + R), via heatmap using R statistical language [40] at pheatmap package [41].**

(bHLH family), NFYA4 and NFYA10 (NFY family), bZIP20 and bZIP45 (bZIP family), GATA11, GATA22, and GATA23 (GATA family), HSFA9, HSFB2-6, and HSFB4 (HSF family) and WRKY1 (WRKY family). Novel eight downregulated cold-responsive TFs were belonged to MYB, NFYC and bZIP TF families including MYB 5, MYB37, MYB38, NFYB3, NFYB4, NFYB9, NFYC2, and bZIP17 in both plants based on the sequence similarity. Some members of TF families showed both up and down-regulation, in addition of being uncommon between the two plants (Table 1 and S2 Table). For instance, *PTL1* and *BMM* from the

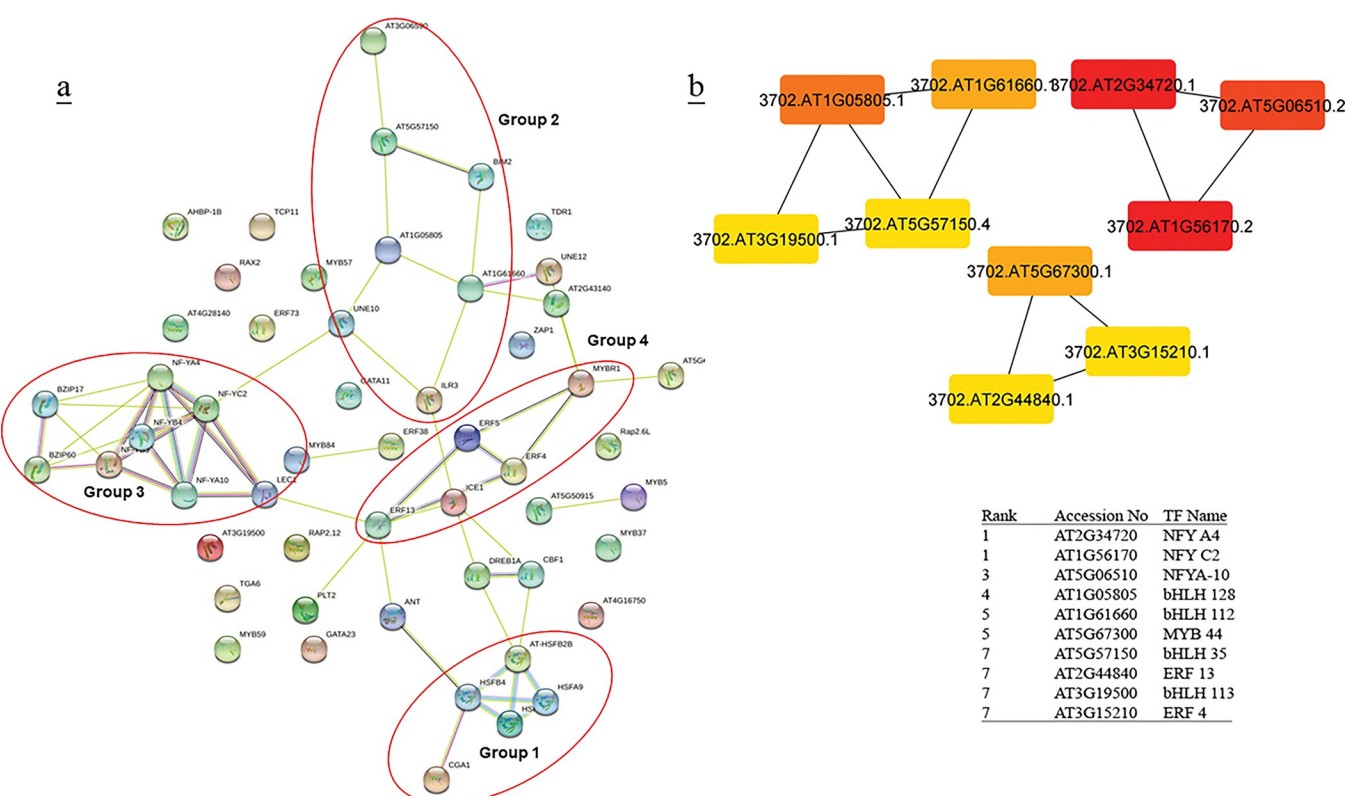

**Fig 2.** (a) Protein-protein interactions of up-and down- regulated TFs in Arabidopsis was created by STRING [51]. Red circles indicate the probable clusters. (b) Hub genes up- and down-regulated TFs in Arabidopsis were identified using Cytoscape [52] through maximal clique centrality (MCC) algorithm. The red nodes represent genes with a high MCC scores, while the yellow node represent genes with a low MCC score.

AP2-like ethylene-responsive TF family showed an increase in expression in rice and a significant decrease in Arabidopsis in response to cold stress. Interestingly, the number of cold-responsive TFs in Arabidopsis was much greater than rice. This finding suggests an evolutionary fact for a plant, *i.e.*, Arabidopsis, that found in a range of geographical latitudes, experiencing varieties of temperatures, versus rice that is a tropical/sub-tropical crop with apparently less-developed regulatory mechanisms in response to cold. For instance, 12 Arabidopsis WRKYs showed up-regulation in Arabidopsis, while only one cold-responsive WRKY was found for rice. Similarly, 22 Arabidopsis NACs showed a significant increase in abundance, whereas rice had no responsive NAC (S2 Table). Most Arabidopsis MYBs showed a significant increase in abundance, while only limited numbers (five) of this family of TFs showed a similar pattern in rice.

For bHLH and G2-like, most corresponding transcripts showed down- and up-regulation in rice and Arabidopsis, respectively. In both plants, NF-Y family members mostly showed a decrease in abundance. In contrast, most TCP and trihelix families, except one in both plants, showed up-regulation that may suggest its importance in cold response. Cold stress increased the abundance of Arabidopsis bZIP significantly. Different members of the MADS-box TF family were responded differently (S2 Table).

## PPI and hub proteins identification

In the analysis of protein–protein interaction of TF families in (A + R), four distinct groups were identified. Arabidopsis groups (G) included ANT and HSF (G1), bHLH members (G2),

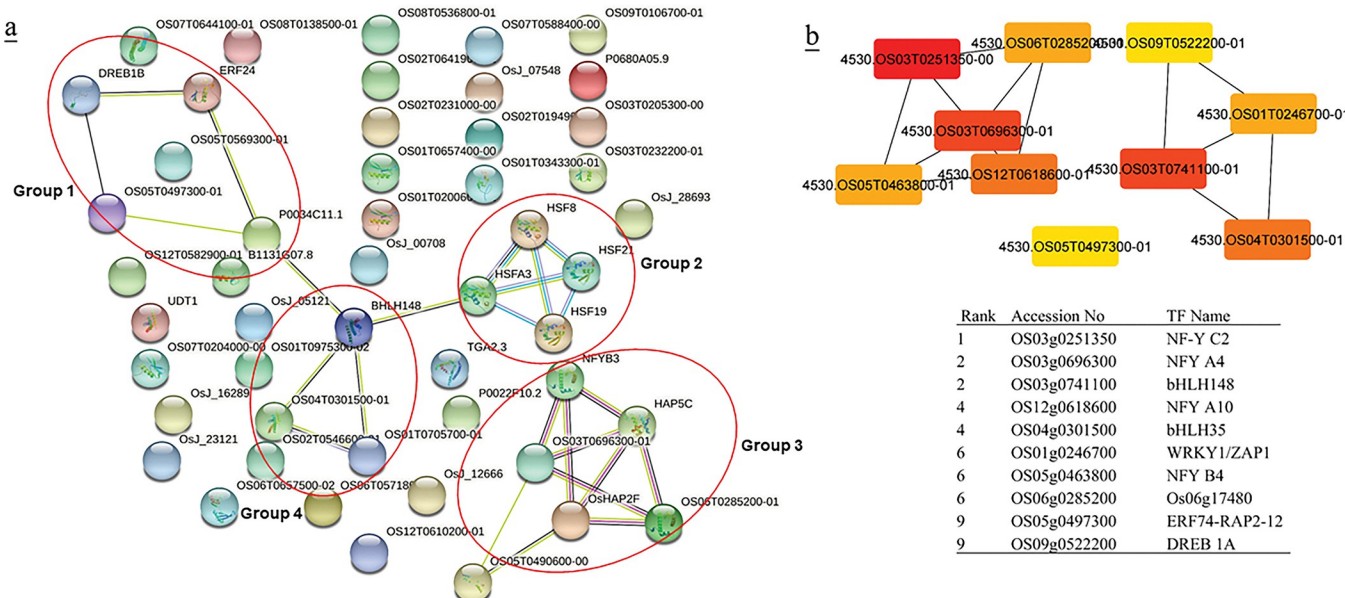

**Fig 3.** (a) Protein-protein interactions of up-and down- regulated TFs in rice was created by STRING [51]. Red circles indicate the probable clusters. (b) Hub genes of up- and down-regulated TFs in rice were identified using Cytoscape [52] through maximal clique centrality (MCC) algorithm. The red nodes represent genes with a high MCC scores, while the yellow node represent genes with a low MCC score.

NFY and bZIP (G3), and ERF and ICE1 (G4) (Fig 2). Co-expressed genes of G1 were mostly active in genetic information processing (folding, sorting and degradation, and transcription) and environmental information processing (signal transduction). Co-expressed genes of G2, G3, and G4 were involved in the biosynthesis of secondary metabolites, carbohydrates, amino acids, lipid, terpenoids, and polyketides metabolism (S13 Table). Rice groups belong to ERF members (G1), HSF members (G2), NFY members (G3), and bHLH members (G4) (Fig 3). Co-expressed genes of G1, G2, and G4 were involved in signal transduction, folding and sorting, terpenoids and polyketides, lipid, carbohydrate metabolism, and biosynthesis of secondary metabolites. Co-expressed genes of G3 were mostly involved in energy metabolism and biosynthesis of secondary metabolites and metabolism of cofactors and vitamins (S14 Table). Significant hub proteins in the TFs protein-protein interaction network were NFY A4 and NFY C2, NFY A10 in both (A + R) (Figs 2 & 3).

PPI network of co-expressed genes of identified TFs and the most significant hub genes showed 1479 and 1283 nodes in rice and Arabidopsis during cold stress, respectively. The most significant hubs in the rice co-expressed genes network were PSI-F, PSI-K, chloroplastic UPF0603, chloroplast photosystem I reaction center subunit, PSI-G and chloroplastic chlorophyll a-b binding protein (Fig 4). In Arabidopsis, the most significant nodes in co-expressed genes network were WRKY40, WRKY33, ZAT10, ZAT12, zinc finger CCCH domain-containing protein 47, zinc finger CCCH domain-containing protein 29, probable CCR4-associated factor 1 homolog 11, probable CCR4-associated factor 1 homolog 11 and calcium-binding protein KRP1 represented (Fig 5).

## Gene ontology of co-expressed genes

Gene Ontology enrichment analysis of co-expressed genes of identified TFs were compared to assess the biological and functional similarity and differences between (A + R), under cold stress (Fig 6, S8 Table). Co-expressed genes in both plants were involved in cellular process,

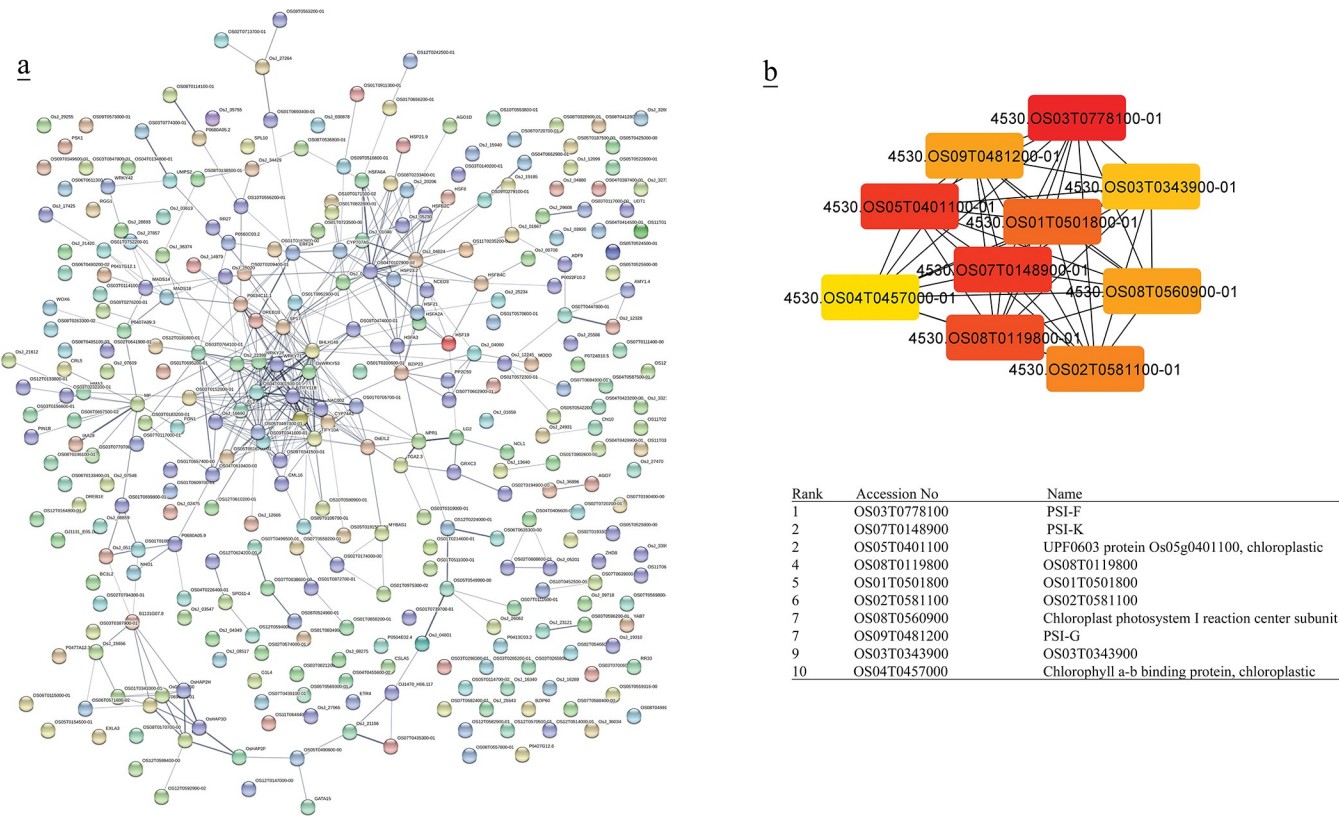

**Fig 4.** (a) Protein-protein interactions of co-expressed genes of up- and down- regulated TFs in rice was created by STRING [51]. (b) Hub genes of co-expressed genes of up- and down-regulated TFs in rice were identified using Cytoscape [52] through maximal clique centrality (MCC) algorithm. The red nodes represent genes with a high MCC scores, while the yellow node represent genes with a low MCC score. Co-expressed genes with MR≤10 were chosen to create PPI network for better visualization.

response to stimulus, signaling, biological regulation, and metabolic process. Co-expressed genes of TFs indicated their involvement in metabolic pathways (lipid and energy metabolism, secondary metabolites biosynthesis), environmental information processing (signal transduction and protein kinase signaling), and environmental adaptation (Circadian clock).

The results indicated that a number of co-expressed genes of identified TFs in response to cold stress, were protein kinases such as serine/threonine-protein kinase and other proteins such as WRKY (a TF) involved in signal transduction with strong reports that these proteins are functioning under the influence of varieties of hormones. For instance, co-expressed genes of the ERF family, active in signal transduction, were under the control of ABA, ethylene and JA. Co-expressed genes of bHLH family of TFs are under the influence of JA, IAA, Auxin, and ethylene. For the NFY family, ABA in rice and SA in Arabidopsis were shown to be effective (S10 Table). In Arabidopsis, kinases are the co-expressed of ERF, MYB, bHLH and bZIP families and in rice with ERF, bHLH, NFY, bZIP, GATA, WRKY, and HSF (S10 Table).

The results showed that 45 co-expressed genes of ERF, MYB, GATA, WRKY, HSF and bHLH TF families in rice and 18 co-expressed genes of Arabidopsis (mostly co-expressed with ERF) are involved in lipid metabolism (*i.e.*, biosynthesis of unsaturated fatty acids α- linolenic acid, linoleic acid, glycolipid, ether lipid; S12 Table).

In rice, 26 co-expressed genes of GATA22 (upregulated) and 31 co-expressed genes of NFYC2 (downregulated) were involved in energy metabolism. *i.e.*, photosynthesis (S14 Table), with six common genes between the two TFs. However, the number of co-expressed genes

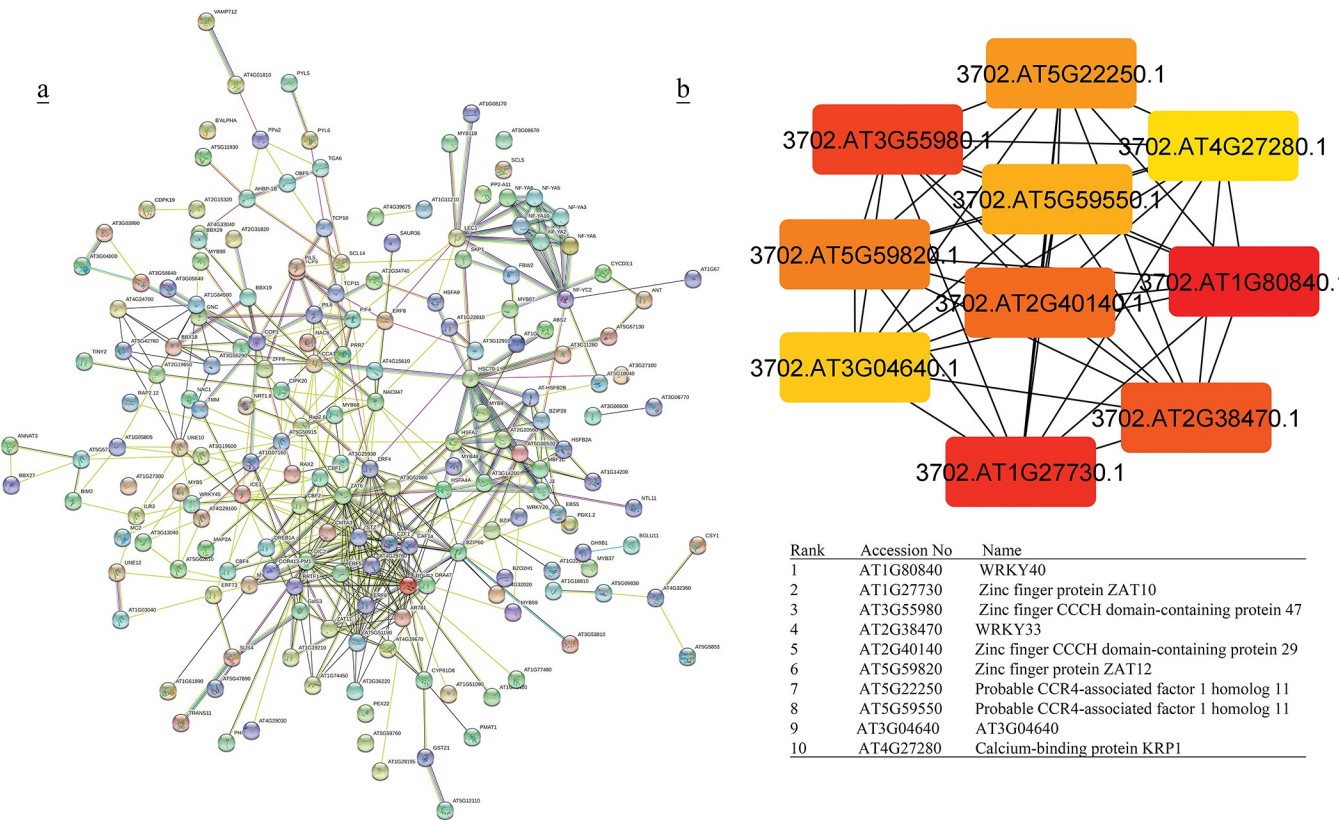

**Fig 5.** (a) Protein-protein interactions of co-expressed genes of up- and down- regulated TFs in Arabidopsis was created by STRING [51]. (b) Hub genes of co-expressed genes of up- and down-regulated TFs in rice were identified using Cytoscape [52] through maximal clique centrality (MCC) algorithm. The red nodes represent genes with a high MCC scores, while the yellow node represent genes with a low MCC scoreCo-expressed genes with MR≤10 were chosen to create PPI network for better visualization.

| Rank | Accession No | Name |
|---|---|---|
| 1 | AT1G80840 | WRKY40 |
| 2 | AT1G27730 | Zinc finger protein ZAT10 |
| 3 | AT3G55980 | Zinc finger CCCH domain-containing protein 47 |
| 4 | AT2G38470 | WRKY33 |
| 5 | AT2G40140 | Zinc finger CCCH domain-containing protein 29 |
| 6 | AT5G59820 | Zinc finger protein ZAT12 |
| 7 | AT5G22250 | Probable CCR4-associated factor 1 homolog 11 |
| 8 | AT5G59550 | Probable CCR4-associated factor 1 homolog 11 |
| 9 | AT3G04640 | AT3G04640 |
| 10 | AT4G27280 | Calcium-binding protein KRP1 |

involved in energy metabolism was fewer in Arabidopsis in contrast to rice, possibly due to the smaller genome size (S13 Table).

The results showed that co-expressed genes of ERF, MYB and bHLH family in Arabidopsis and co-expressed genes of bHLH, GATA and NFYB and MYB families in rice were involved in the metabolism of secondary metabolites including sugar and amino acid osmolytes (S13 & S14 Tables). In rice, co-expressed genes of ERF (cryptochrome 2, Che Y-like domain containing protein), HSF (Che Y-like domain containing protein, FKF1, GIGANTEA protein) and bZIP (phytochrome A) TFs were involved in circadian rhythm (S13 and S14 Tables). Co-expressed genes of ERF (cryptochrome 2), bHLH (HY5 TF), NFYA (CCA1), bZIP (SPA1-related 2) and WRKY (E3 ubiquitin-protein ligase RFWD2) were involved in circadian rhythm in Arabidopsis (S13 and S14 Tables).

## TF interactions

Our data were suggestive of the interaction of TFs with other TFs or proteins (S5 Table). For example, DREB1B with DREB1A and CBF4; MYB57 with MYB21 and MYB27; NFYA-4 (upregulated TF) with NFYB3 (downregulated TF) and NFYA7 (S4 Table). Our data showed that TF interactions could also happen between different types of TF families such as ERF4 and bHLH, MYB59, and GATA/NAC20 (S5 Table; Fig 7). In order to identify the significant hub proteins having interaction with identified TFs, a PPI network was constructed. The most

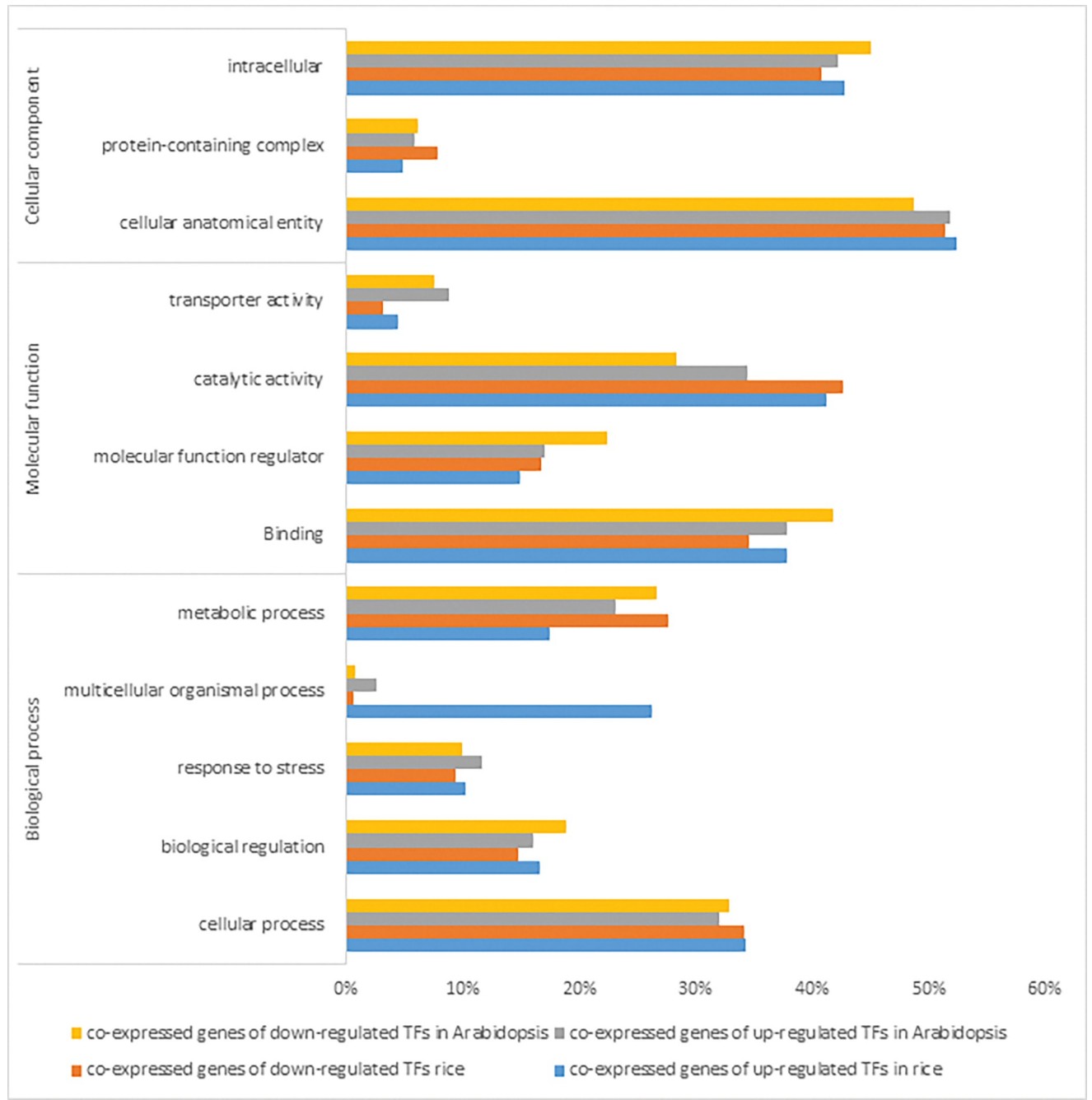

**Fig 6. Gene ontology results of co-expressed genes of identified up-and down- regulated TFs in (A + R) in response to cold stress was created using PANThER [45].** Annotations lower than 2% are not presented.

significant proteins with high rank interaction with TFs were UNE12 (AT4G02590) and NPR1 (AT4G19660), bHLH (AT1G03040), TGA-bzip (AT5G06950), NF-YB3 (AT4G14540), and bHLH105/ILR3 (AT5G54680) (Fig 7). Furthermore, four members of the ERF family (DREB 1A, DREB 1B/CBF1, ERF 4, ERF 113), three members of the bHLH family (bHLH148, BIM2, UNE10), two members of the bZIP family (bZIP20, bZIP45) and MYB59 have indirect interaction with NPR1. Here, bHLH105/ILR3 TF showed to be in direct interaction with KNAT7

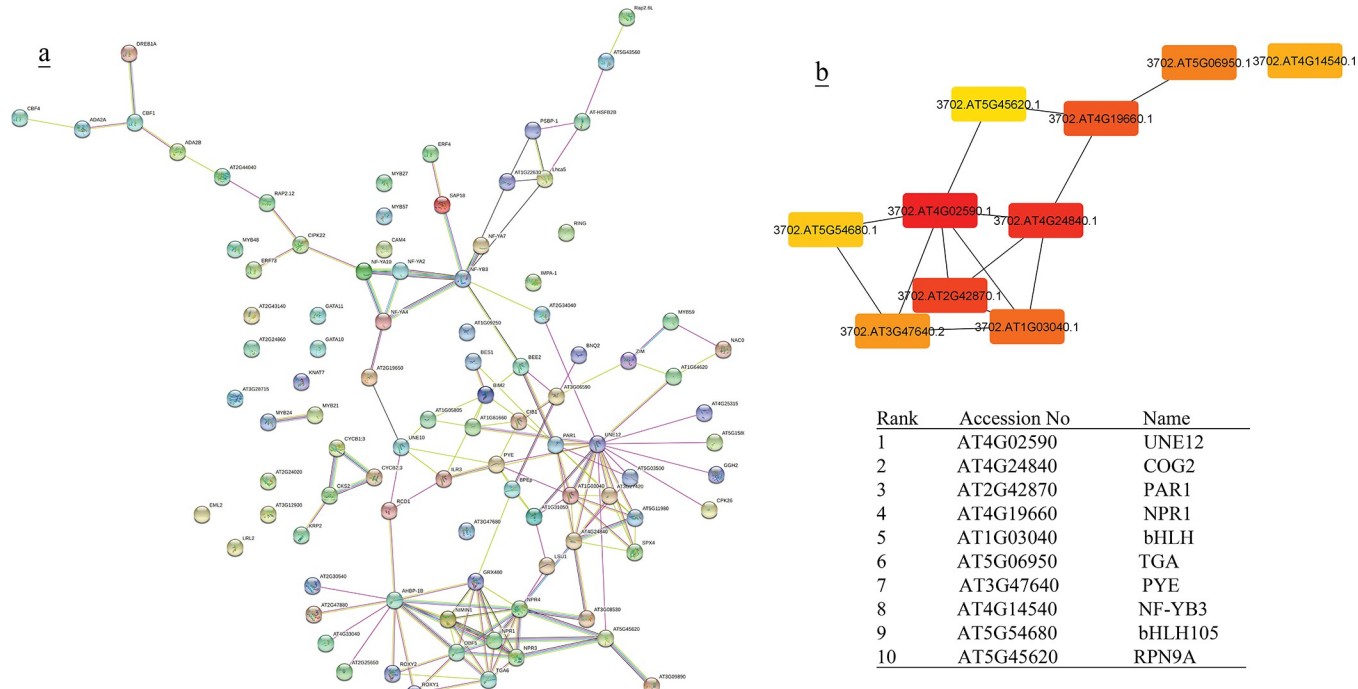

| Rank | Accession No | Name |
|------|-------------|------|
| 1 | AT4G02590 | UNE12 |
| 2 | AT4G24840 | COG2 |
| 3 | AT2G42870 | PAR1 |
| 4 | AT4G19660 | NPR1 |
| 5 | AT1G03040 | bHLH |
| 6 | AT5G06950 | TGA |
| 7 | AT3G47640 | PYE |
| 8 | AT4G14540 | NF-YB3 |
| 9 | AT5G54680 | bHLH105 |
| 10 | AT5G45620 | RPN9A |

**Fig 7.** (a) The PPI network shows the up- and down- regulated TF interactions with other TFs or proteins. TF regulatory interactions were retrieved from CORNET [34] using both experimental and predicted data of IntACt, TAIR and AtPID regulatory interactions. Edges represent the protein-protein associations. The red nodes represent genes with a high MCC scores, while the yellow node represent genes with a low MCC score.

(Homeobox protein knotted-1-like 7). bHLH116 (ICE1) had an interaction with MYB15. In downregulated TFs, bHLH12 and NFYB-3 had an interaction with bHLH59/UNE12 and NFYA4, respectively.

## miRNA

psRNATarget was used for common TFs and their probable interactive miRNAs. Upregulated TFs were predicted to be targeted with 121 and 38 miRNAs in rice and Arabidopsis, respectively. The number of predicted miRNAs for downregulated TFs were 74 and 18 in rice and Arabidopsis, respectively. Probable significant interactive miRNAs were chosen based on minimum free energy hybridization of mRNA-miRNA with a threshold of Mfe $\leq$ -30 kcal/mol for each up-and down-regulated TFs (Table 2 and S6 Table). In rice, miR5075, miR156, miR2927, miR159a.2, and miR1846 target 29, 13, 12, 12, and 10 TFs from different families such as ERF, bZIP, bHLH, HSF, and NFY (Table 2). In Arabidopsis, miR395 and miR858, and miR414 target 12, 8, and 5 TFs. Some individual miRNAs were predicted to target both TFs with an increase and decrease in their abundance in Arabidopsis including miR414, miR391, miR834, and miR771, and in rice miR351, miR5833, miR2925, and miR5075. In Arabidopsis, miR5658 targeted only downregulated TFs including PTL2, MYB38, and TCP1. TCP21 and MYB38, downregulated TFs, are being regulated with "miR5658" in Arabidopsis and with "miR2925" in rice. According to the number of mismatches found, it seems that translation halt is the mode of action to control TF formations in both model plants (S6 Table). Six TFs in both plants were predicted to be targeted by only one specific miRNA in each plant, including ANT (ath-miR5020c, osa-miR6255), ERF5 (ath-miR414, osa-miR437), bHLH35 (ath-miR1886.1,

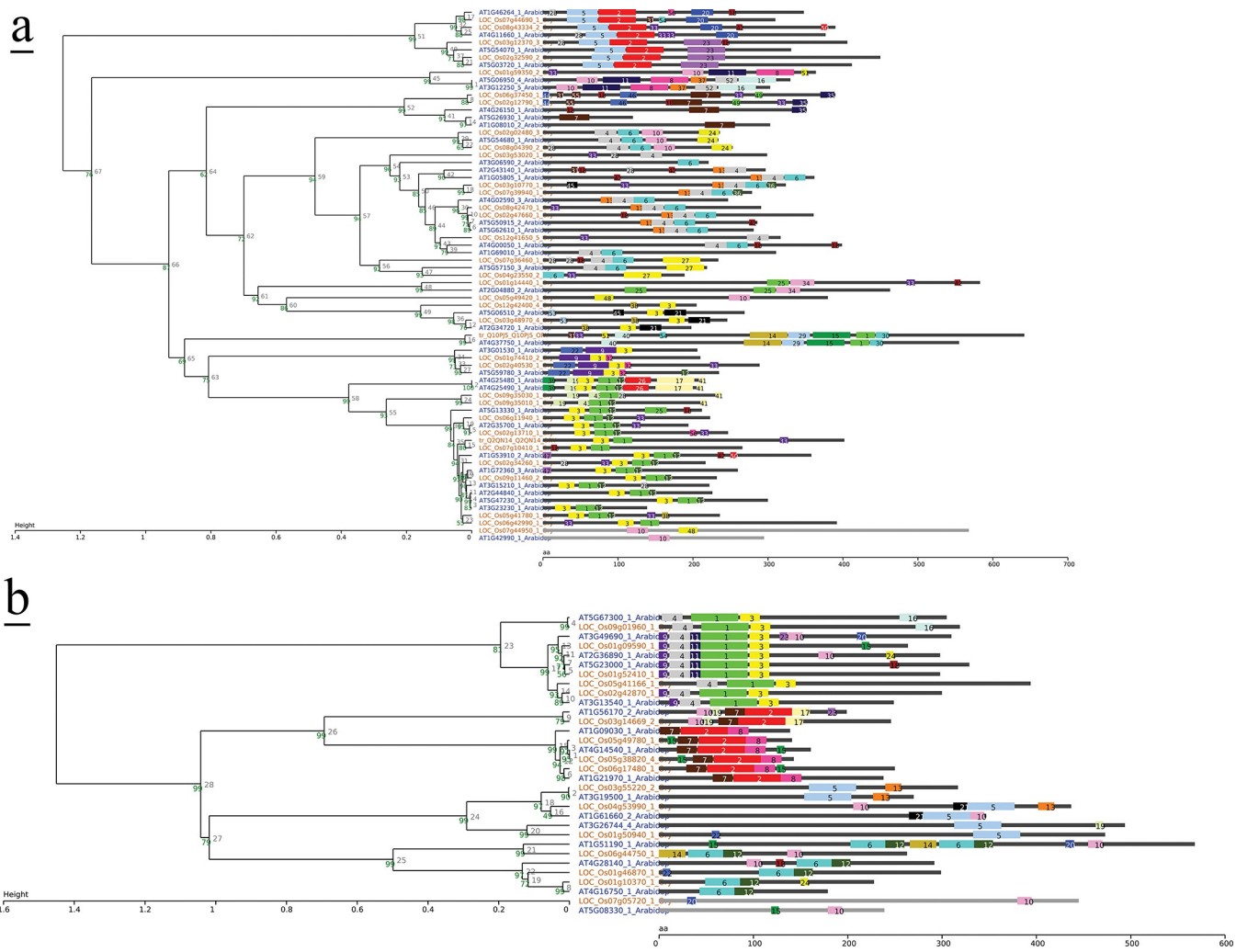

**Fig 8. a**: Conserved domain analysis of TF genes with increased abundance in (A + R) using SALAD [31]. **b**: Conserved domain analysis of TF genes with reduced abundance in (A + R) using SALAD [31].

osa-miR414), bHLH137 (ath-miR5023, osa-miR5515), NFYA10 (ath-miR836, *osa-miR2873a)*, and GATA23 (ath-miR5020b, osa-miR168b) (Table 2).

## TF sequence conservation and promoter analysis

The analysis of conserved domains in all common up-and downregulated TFs of rice and Arabidopsis was performed at SALAD (Fig 8A and 8B) to determine DNA-binding domains (DBDs) of each TF family. The number of DBDs in all identified TFs in both plants were the same, except for WRKY1 and PLT2 (S3 Table). Sequence homology search of each domain supports the conservation (S3 Table).

Database review results indicated that each TF family is under the control of a specific phytohormone. For example, the ERF family is controlled and induced by ABA, SA, ethylene and Auxin (S7 Table), while the MYB family are under the influence of jasmonic acid in addition to the same hormones (S7 Table). Promoter analysis of putative cold-responsive TFs showed that phytohormone responsive *cis*-elements such as ABRE (ABA-responsive elements), TGA-

element (auxin-responsive element), TCA-element (*cis*-acting element involved in SA respon-
siveness) and LTR (low-temperature responsive elements) are common.

To seek if the co-expressed genes to TFs found here are true, promoter analysis was carried
out on protein kinases as one of the main players in cold response. Interestingly, the results
were indicative of the presence of *cis*-elements in protein kinase genes that allows the recogni-
tion and binding by the identified TFs.

## Discussion

Investigations aimed at deciphering the molecular events that underpin the initiation and pro-
gression of abiotic stress response in plants are primarily targeted towards TFs, whose expres-
sion, contributes to alterations in molecular function and lead to stress response through the
regulation of downstream networks. Bioinformatics and *in silico* analysis of biological systems
could provide some predictions that might come true in laboratory [55]. As it has practiced in
rice [55–57] and Arabidopsis [58, 59].

Here, we attempted to pinpoint to the regulatory network of TFs and microRNAs that might
be involved in cold stress response in a comparative approach between the two model plants, *i.
e.*, (A + R). According to PlantTFdb, the identified common TF genes in this study include
2.30% and 2.20% of the total number of 2296 and 2408 genes in (A + R), respectively [32].

### Analysis of PPI networks

In this study, PPI network analysis showed that some TFs, *i.e.*, NFYA4, NFYA10, and NFYC2,
could be considered as the hub genes in both plants (Figs 2 & 3). Nuclear Factor-Y (NF-Y),
plays an important role at various stages of plant growth and development, especially in
response to stress [60]. Here, according to the analysis of TF co-expressed genes in rice, NFY
TFs have regulatory effects on energy metabolism and biosynthesis of secondary metabolites
(Fig 3, S14 Table), while they have roles in biosynthesis of secondary metabolites and carbohy-
drates metabolism with fewer genes in Arabidopsis. This might be due to the smaller genome
size of Arabidopsis (Fig 2, S13 Table). Previous studies revealed that NF-Y members were
involved in the stress response. In rice, OsNF-YA1 was downregulated under both drought
and cold stress and OsNF-YA5 was downregulated in response to cold treatment [61]. *Arabi-
dopsis* NF-Y has an important role in the responses to abiotic stresses [62]. Kreps et al. [63]
identified Arabidopsis NF-YB2 through microarray analysis to be upregulated by NaCl, man-
nitol, or cold (4˚C). Hackenberg et al. [64], reported the transcript level of AtNF-YC2 was
highly induced by light, oxidative, heat, cold, and drought stress, while NF-YC4 was induced
by cold. NF-YB2 expression in *A. thaliana* seedlings (16 day-old) was downregulated during
early (0.5, 1 and 3 h) cold stress response and upregulated at the later stages (6, 12 and 24 h).
Similar switching behaviour was displayed for AtNF-YB4 and AtNF-YB8, revealing these
genes to play a putative role in the late stages of plant adaptation to cold [65].

### TFs target metabolic pathways

The comparison of the co-expressed gene network and their hubs in (A + R) indicated partici-
pation in different metabolic pathways. The most significant hubs in the rice co-expressed
gene network of upregulated TFs were PSI-F, PSI-K, chloroplastic UPF0603, chloroplast pho-
tosystem I reaction center subunit, PSI-G and chloroplastic chlorophyll a-b binding protein
(Fig 4). On the other hand, PPI and gene ontology of data showed that most of the co-
expressed genes of cold-induced TFs in rice were involved in energy metabolism, lipid metab-
olism, biosynthesis of secondary metabolites, folding, sorting and degradation and transcrip-
tion, terpenoids and polyketides metabolism, and circadian rhythm. However, the most

significant hubs in the Arabidopsis co-expressed gene network were WRKY40, WRKY33, ZAT10, ZAT12 (Fig 5), which have been reported as TFs involved in cold stress. These results suggest that rice is more engaged in energy metabolism especially photosynthesis during cold stress (S13 and S14 Tables). Low temperature severely affects the growth and development of plants, especially photosynthesis [66]. Chloroplasts can also perceive chilling stress signals via membranes and photoreceptors, and they maintain their homeostasis and promote photosynthesis by regulating the state of lipid membranes, the abundance of photosynthesis-related proteins, the activity of enzymes, the redox state, and the balance of hormones by releasing retrograde signals, thus improving plant resistance to low temperatures [66].

Here, lipid metabolism in rice seems to have a greater role in response to cold treatment in comparison to Arabidopsis (S12 Table). Lipid metabolism and remodeling, which modulate the lipid composition, fatty acyl group unsaturation, and membrane fluidity [67], is essential to plant cold tolerance [68]. Genome-wide association mapping of cold tolerance in cultivated rice revealed 87 cold tolerance-related quantitative trait loci (QTLs) with significant enrichment in lipid metabolism [69].

Our data was demonstrated the role of protein kinases and their cross-talk with phytohormones in cold-induced signal transduction (S10 Table). Different Protein kinases were detected in cold response in (A + R) such as MAP (Mitogen-Activated Protein) Kinase and LRR receptor-like serine/threonine-protein kinase.

The role of these protein kinases in cold response has been studied in rice (*Oryza sativa*) and *A. thaliana*. MAP Kinases are a class of protein kinases that play important roles in signal transduction pathways, including those involved in plant responses to various stresses such as cold stress [70]. Studies have shown that MAP Kinases play a crucial role in the cold response pathway. In rice, activation of MAP Kinases upon exposure to cold stress leads to the phosphorylation of downstream target proteins, which trigger various cellular and molecular responses, such as changes in gene expression, accumulation of osmoprotectants, and modulation of ion transporters to cope with cold stress [71]. MAP Kinases in rice have been found to interact with other cold-responsive proteins and TFs, forming a complex regulatory network that modulates the plant's response to cold stress [72]. Cold stress activates MAP Kinases in Arabidopsis and regulates downstream targets, leading to changes in gene expression and various physiological responses, such as alterations in lipid metabolism, accumulation of osmoprotectants, and induction of antioxidant defense mechanisms [73].

The other protein kinases, LRR receptor-like kinases, are a type of receptor proteins that play a key role in many abiotic stress and physiological processes such as regulating gene expression responses and sensing external signals at the cellular level [74]. In rice, the expression of OsLRR2 in the leaves at the seedings, booting and flowering stages were markedly upregulated after cold and drought treatment [74]. The COLD1 (COLD REGULATED 1), a LRR receptor-like kinase in Arabidopsis, was shown to play a crucial role in cold perception and signalling. COLD1 regulates the expression of C-repeat binding factors (CBFs), which are key TFs involved in cold response, leading to changes in gene expression and cold tolerance in Arabidopsis [75].

The other important finding was the crucial role of circadian rhythm with greater numbers of TFs in Arabidopsis than rice (S13 and S14 Tables). One such TF was CBF, as the core component of circadian clock, reported to be induced by cold [76].

In rice, co-expressed genes of downregulated TFs were involved in metabolic pathways such as biosynthesis of secondary metabolites (46 genes), energy metabolism (31 genes), lipid (22 genes), amino acid metabolism (20 genes), signal transduction (17 genes) and carbohydrate metabolism (10 genes) (S1 Fig). In Arabidopsis co-expressed genes of downregulated TFs were involved in biosynthesis of secondary metabolites (21 genes), signal transduction (11 genes) and carbohydrate metabolism (10 genes) (S1 Fig).

## TF interactions

The investigation of proteins interacting with TFs is of great importance. It has been shown that TFs interact with other TFs to form functional protein complexes [77]. Additionally, kinases may interact with TFs to act as a molecular switch to toggle their activities via phosphorylation [78], and many TFs form functional complexes such as some NAC and MADS TFs, which form homo- or hetero-dimeric or tetrameric complexes [79]. Combinatorial interactions between TFs are important for the regulation of downstream genes [80]. In this study, indirect interaction between bHLH59 with KNAT7 (Homeobox protein knotted-1-like 7) was pointed, indicating potential cross-family interactions between different types of TFs. The *KNAT7* is a Class II KNOTTED1-like homeobox (KNOX2) TF that acts as a negative regulator of secondary cell wall biosynthesis in inter fascicular fibres [80].

The cell wall is clearly affected by many abiotic stress conditions. A common plant response is the production of ROS and an increase in the activity of peroxidases, XTH (xyloglucan endotransglucosylases/hydrolases) and expansins [81]. KNAT7 forms a functional complex with OFP proteins to regulate aspects of secondary cell wall formation and OFP6 confers resistance to drought and cold stress in plants including rice [82]. Li et al. (2011) [83] proposed that KNAT7 forms a functional complex with OFP proteins to regulate aspects of secondary cell wall formation. They reported that AtOFP1 and AtOFP4 are components of a putative multi-protein transcription regulatory complex containing BLH6 and KNAT7. Accordingly, our data revealed that TF interactions may occur, suggesting potential cross-talk regulatory mechanisms in transcriptional regulation.

The other interesting example of such cross-family interactions is the indirect interaction between NPR1 (Non-expressor of Pathogenesis-Related Genes 1), a transcription co-activator involved in plant defense responses [84] and several TFs including four members of ERF family (DREB 1A, DREB 1B/CBF1, ERF 4, ERF 113), three members of bHLH family (bHLH148, BIM2, UNE10) and MYB59 (S5 Table). NPR1 is an essential regulator of plant systemic acquired resistance (SAR), which confers immunity to a wide spectrum of pathogens [85]. Singh et al. [86] reported that 7 days of repetitive cold stress (1.5 h at 4°C on day 1) activated the pattern-triggered immunity in Arabidopsis. Similarly, Kim et al. [87] detected increased disease resistance in 3 weeks of cold stressed Arabidopsis plants, indicating NPR1 is partially required for cold activation of disease resistance, and there exists an NPR1-independent SA pathway in cold activated immunity. It is suggested that the short-term cold stress can act as a priming stimulus to prime defence response of Arabidopsis to bacterial pathogens [88]. Taking these notions into account, it could be concluded that there is a crosstalk between cold stress and immunity. The results of our study indicate that while TFs generally tend to interact with TFs from their own family, it does not mean that interactions with other families should be ignored.

## miRNA

The abiotic stress response network mediated by miRNA is important in plant response to various stresses [89]. Here, TF-miRNA interactions seem to be different in (A + R) in terms of number of miRNA and mode of action in response to cold stress (Table 2). According to the results miR5075 targets most TFs in rice, while TFs in Arabidopsis are regulated by diverse sets of miRNA (Table 2). We found that the numbers of responsive miRNAs to cold stress in rice were greater than Arabidopsis. According to the results miR5075 targets most TFs in rice, while TFs in Arabidopsis are regulated by diverse sets of miRNA (Table 2). In addition, translation halt was the preferred mode of action in the post-transcriptional regulation mechanism in both plants.

TFs and microRNAs play important roles in regulating the activity of the genes at transcriptional and post-transcriptional levels [90, 91]. Under cold stress, variations in miRNAs expression (either up- or down-regulation) modify the transcript abundance of their target genes [92, 93]. For example, overexpression of rice miRNA156 was resulted in an increase in cell viability and growth rate under cold stress in rice and other plants through targeting OsSPL3 and other TFs [94]. According to their targets, miRNAs respond to low temperature stress through three tactics: the first is respond to abiotic stress directly; the second is indirectly responding to external stimuli by regulating TFs that relate to stress responses; and the third is that miRNAs can respond to multiple stresses and their target genes could code certain hydrolases or oxidoreductases [95].

In this study, 192 new miRNA targeting up- and down-regulated TFs were identified in rice. Some of the novel miRNAs in relation to cold stress were miR5075, miR2927, miR159a.2 and miR1846 (Table 2). Our findings were also in accordance with earlier studies. For instance, miR319 (reported by [96]) targets ERF38, bHLH79, MYB5, and TCP1. miR398b [97] targets ERF74 and miR528 [98] targets ERF73, DREB1B, and GATA22 (Table 2). Tang and Thompson (2019) [98] demonstrated that the overexpression of rice miRNA528 increased cell viability, growth rate, antioxidants content, ascorbate peroxidase (APOX) activity, and superoxide dismutase (SOD) activity under low-temperature stress in (A + R). Their results suggested that OsmiR528 increases low-temperature tolerance by modulating the expression of the corresponding TFs.

miRNAs regulate at post-transcriptional level. Since some of the target genes are TFs, plant miRNAs have emerged as the promising targets for crop improvement, specifically at stress response conditions as they control intricate agronomic traits [99, 100]. In Arabidopsis, TF-miRNA interactions have been implicated in the regulation of cold stress responses. For example, over-expression of miR402 improved tolerance to salinity, drought, and cold stress in [101]. Here, it was found that the number of reported miRNAs for cold stress in Arabidopsis were greater than rice; some of which has already reported in response to cold stress. For instance, some of the previously reported miRNAs were miR156 [102], miR165, miR168, miR169, miR171, miR172, miR319, miR393, miR396, miR397 [103], miR402 [104], miR408 [105], miR157, miR159, miR164, miR166, miR394, miR398 [106], miR394a [107], miR397a [108], miR402 [109]. Identified miRNAs in Arabidopsis, miR157, miR171, miR393, and miR396 were in accordance with the literature which in this study target NFYB3, MYB5, ERF98, and ERF74, respectively. Also, miR156 targets bHLH116/ICE1 and HSFA3.

These investigations prove the importance of miRNAs for plants. Overexpression or repression of the miRNAs could lead to down-/up- regulation of their downstream target genes. Consequently, resulting in pleiotropic phenotypes in plants [110]. The studies based on stress-response has shown that miRNAs can provide important understanding in plant stress resistance breeding and gene expression. Thus, comprehensive understanding and application thereof can be harnessed as a powerful novel approach in the development of adaptive mechanism in plants [110].

## Promoter analysis

Promoter analysis of cold-responsive TFs revealed the presence of *cis-elements* that can be induced/regulated by phytohormones (S9 and S11 Tables). Moreover, promoter analysis of protein kinase co-expressed genes revealed the existence of *cis*-element binding sites of identified TFs (S10 Table). The role of hormonal signaling in response to cold stress is unequivocal in the activation process of TFs [111, 112]. Cold-stress response gene-expression pathways are classified as ABA dependent or ABA independent. ABF/AREB-dependent gene-expression

pathways are ABA dependent, DREB1/ CBF-dependent cold-response gene-expression pathways are ABA independent, and the two pathways are cross-linked and interdependent [113]. ABA, GA, brassinosteroids (BR), JAs, auxin, cytokinin (CK), melatonin, and polyamines have been reported as the major hormones and growth factors contributing to cold stress response [114]. For instance, ABA surge in cold stress leads to higher expressions of the *CBF* genes possibly via binding to the CRT/DRE elements [115]. Moreover, ethylene, ABA, and JAs can induce the expression of *ethylene-responsive* (*ERF*) genes [116]. GA and JA play trivial roles in the ICE-CBF-COR pathway [113]. As it became evident, phytohormone responsive *cis*-elements were the most common elements in promoters of the corresponding TF genes in this study.

## Conclusion

We compared common up- and down- regulated TFs in (A + R) in response to cold stress to provide a detailed sense of the pathways and candidate TFs. The potential target genes of cold-responsive TFs were detected through co-expression network to uncover the regulatory networks involved in cold stress in (A + R). The construction of regulatory networks of TFs provides a comprehensive view of the molecular mechanisms underlying cold stress response. The results showed a significantly different regulatory mechanism of each TF in each plant in terms of co-expressed genes, interacting partners, downstream regulatory networks and pathways. In rice, the most significant hub genes were involved in photosynthesis, and in Arabidopsis they were the TFs involved in signal transduction and biosynthesis of secondary metabolites, suggesting that rice is more engaged in energy metabolism in contrast to Arabidopsis in response to cold. These finding have merits for further experimental analysis. Presented TFs, miRNAs and co-expressed genes in this study should be validated in terms of regulatory interactions between cold-responsive TFs and their target genes to confirm the functional relevance of the predicted regulatory networks. Knowledge about the regulatory networks of genes and proteins that define the cold-stress response is important in concepts of evolutionary biology among genera, helpful in defining subtle differences present within a species in response to varieties of stresses, and ultimately helpful towards the engineering of resilient plants before cold stress. Comparative transcriptional studies could also be used as a framework to investigate the regulatory networks of biotic and abiotic stress responsive TFs in various plant species to contribute the advancement of plant stress biology research.

## Supporting information

**S1 Table. Accession number of *A. thaliana* and *O. sativa* Two-week old seedlings microarray from GEO.**
(DOCX)

**S2 Table. Uncommon up- and down-regulated TFs in rice and Arabidopsis.**
(DOCX)

**S3 Table. Transcription factor specifications obtained from plant transcription factor database (http://planttfdb.gao-lab.org/).**
(DOCX)

**S4 Table. Protein BLAST [33] results of common up- and down-regulated TF genes in Arabidopsis and rice.**
(DOCX)

**S5 Table. TF interactions with other TFs or proteins were obtained from CORNET [34] based on IntACt, TAIR and AtPID databases.** The metabolic pathways of each interacted protein were obtained from KEGG Pathway database [50].
(DOCX)

**S6 Table. Predicted microRNA-TF in Arabidopsis- hybrid position, graph and mfe obtained using psRNATarget [37] and RNAhybrid [38].**
(DOCX)

**S7 Table. Phytohormonal control of TFs obtained from Plant TFDB [32].**
(DOCX)

**S8 Table. Gene ontology results of co-expressed genes of up- and down-regulated TFs in rice and Arabidopsis using PANTHER [45].**
(DOCX)

**S9 Table. Phytohormone- and abiotic stress- related *Cis*- elements in promoter regions of rice "R" and Arabidopsis "A" TFs.**
(DOCX)

**S10 Table. Promoter analysis of co-expressed protein kinases genes of each TF in two model plants were obtained from PlantPAN [48] and AGRIS [49].**
(DOCX)

**S11 Table. Co-expressed genes of TF families in rice and Arabidopsis were active in signal transduction under the control of different hormones.** Data was obtained from KEGG [50].
(DOCX)

**S12 Table. Co-expressed genes of putative cold-responsive TFs in rice and Arabidopsis involved in lipid metabolism.**
(DOCX)

**S13 Table. Metabolic pathways that co-expressed genes of Arabidopsis TFs are involved in each group.**
(DOCX)

**S14 Table. Metabolic pathways which co-expressed genes of rice TFs are involved in each group.**
(DOCX)

**S1 Fig. Metabolic pathway of co-expressed genes of down-regulated TFs in rice and Arabidopsis.**
(DOCX)

**S1 Graphical abstract. The gene expression data of eight cold-treated microarray datasets were retrieved from GEO for *A. thaliana* and *O. sativa* in seedling stage treated for 24 h at 0–5˚C.** Common TFs were separated. *In silico* analysis was applied including conserved domain analysis, TF interactions, post-transcriptional analysis (miRNAs), co-expression analysis, gene ontology, and PPI Network.
(TIF)

## Author Contributions

**Conceptualization:** Naser Farrokhi.

**Data curation:** Khazar Edrisi Maryan.

**Formal analysis:** Khazar Edrisi Maryan.

**Investigation:** Khazar Edrisi Maryan.

**Methodology:** Naser Farrokhi.

**Project administration:** Naser Farrokhi.

**Supervision:** Naser Farrokhi.

**Writing – original draft:** Khazar Edrisi Maryan.

**Writing – review & editing:** Naser Farrokhi, Habibollah Samizadeh Lahiji.

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
