## [Decision Letter · Decision Letter 0]

22 Mar 2023

PONE-D-22-30309Cold-responsive transcription factors in Arabidopsis and rice: A regulatory network analysis using array data and gene co-expression networkPLOS ONE

Dear Dr. Farrokhi,

Thank you for submitting your manuscript to PLOS ONE. After careful consideration, we feel that it has merit but does not fully meet PLOS ONE’s publication criteria as it currently stands. Therefore, we invite you to submit a revised version of the manuscript that addresses the points raised during the review process. 

We look forward to receiving your revised manuscript.

Kind regards,

Keqiang Wu, Ph.D

Academic Editor

PLOS ONE

Journal Requirements:

Reviewers' comments:

Reviewer's Responses to Questions

**Comments to the Author**

1. Is the manuscript technically sound, and do the data support the conclusions?

Reviewer #1: Partly

Reviewer #2: Partly

Reviewer #3: Yes

2. Has the statistical analysis been performed appropriately and rigorously? 

Reviewer #1: Yes

Reviewer #2: Yes

Reviewer #3: Yes

3. Have the authors made all data underlying the findings in their manuscript fully available?

Reviewer #1: Yes

Reviewer #2: Yes

Reviewer #3: Yes

4. Is the manuscript presented in an intelligible fashion and written in standard English?

Reviewer #1: Yes

Reviewer #2: Yes

Reviewer #3: Yes

5. Review Comments to the Author

Reviewer #1: This manuscript deals with the exploration of cold-responsive transcription factors in Arabidopsis and rice. These results were preliminarily inferred ones which needs to be further analyzed for deriving sound concepts. The text seems mainly descriptive some TF and miRNA genes while the major scientific question to be answered is to be clarified. The authors should mention clearly the major scientific findings or progresses in comparison to the previously reported results.

Abstract

*Line 11-14 in Page 2. A clear conclusion is missed in the abstract. Please make definitive conclusions what authors really achieved from their results.

Introduction

*Line 4 in Page 5. The common and specific mechanisms of cold tolerance in rice and Arabidopsis should be elaborated.

*Lines 11-13 in Page 5. Authors have to cite examples on the effect of miRNA in cold resistance. Authors should cite some literature about the interaction mechanism of miRNA and mRNA in cold response.

Methods

*Line 5-7 in Page 6. What are the criteria for selecting a DET? Was it detected simultaneously by the 8 GO datasets or by any one of them?

Results

*Line 13 in Page 10. As shown in Figure 6, what are the differences between the GO terms for down- and up-regulation TFs? Most of them do not show a clear distinction between the two, so could down- and up-regulated TFs be combined to perform GO analysis?

*Line 16 in Page 10, Lines 1-2 in Page 11.The pathway results should be provided in the Supplementary Table.

*Line 9 in Page 12. How does Figure 7 differ from Figure 2 in terms of the construction method? Authors need to provide additional details.

*Line 6-8 in Page 14. The results should be provided to illustrate this conclusion.

Discussion

*Line 1-2 in Page 15. Authors have mentioned that "In this study, PPI network analysis showed that some TFs, i.e., NFYA4, NFYA10, and NFYC2, could be considered as the hub genes in both plants. Combined with the previous studies, discuss the results with respect to the roles of these hub genes in cold stress.

*Line 14 in Page 15. A comparison of metabolic pathways (and hub genes) between rice and Arabidopsis should be elucidated.

*Line 8-14 in Page 16. Based on your findings, please discuss the role of detected kinase in cold response rather than describing the conclusions of previous studies.

*Line 1-4 in Page 17. Authors described too many results in the discussion section with few references and little analysis. Authors should discuss the results with the appropriate literature.

*Line 7-10 in Page 18. “TF-miRNA interactions seem to be different in Arabidopsis and rice in response to cold stress”. Discuss the results with respect to the role of TF-miRNA interactions in cold stress.

*Line 3 in Page 20. A summary should be derived from the above analysis in the perspective of regulatory networks. Besides, Authors have to add the future concept of the study.

Reviewer #2: The article entitled "Cold-responsive transcription factors in Arabidopsis and rice: A regulatory network analysis using array data and gene co-expression network" has chosen an important abiotic stress (cold) for investigation. Some comments are suggested to improve the current version of this manuscript.

1. The description of materials and method needs to be revised. The descriptions of the data do not match the relevant tables completely. For example, the gene expression data of eight cold-treated microarray datasets (GEO) presented in table S1 are not only in the conditions of 4-5 °C, and 0 °C are also seen in these data. Therefore, different conditions may have different effects on the result of gene expression.

2. It is better to provide correct and more complete explanations for the figures and tables of the manuscript.

3. Which the authors claim in this report, new TFs, miRNAs and co-expressed genes have been introduced as cold-responsive markers, also the authors claim these cold-responsive markers can be used in future studies and the development of tolerant varieties. It would have been better to add a verification analysis or some kind of confirmation to this article. Because the number of introduced genes, TFs, miRNAs is large and it is necessary to limit them in a way and to introduce cold-responsive markers. Especially, it is likely that what was introduced in this research is not specific to the conditions of cold stress and may have a different expression in other stresses, especially in abiotic stresses. Therefore, it is better to investigate and report the expression and behavior of introduced TFs, miRNAs and genes in other abiotic stresses such as drought and heat. If there are common in abiotic stresses, it is necessary to identify them, and according to the title of the article, cold-responsive transcription factors in Arabidopsis and rice should be specifically introduced.

Reviewer #3: This study provides a comparative analysis of the transcriptional regulatory response to cold stress in rice and Arabidopsis, with a focus on the identification of up- and down-regulated TFs and miRNAs. The results show differences in the number and diversity of TF families in each plant, as well as differences in the regulatory mechanisms of each TF. Additionally, miRNAs in Arabidopsis were found to target TFs more specifically compared to rice. The study highlights the importance of understanding the regulatory networks involved in the response to cold stress in plants, and provides a basis for further experimental analysis and the engineering of resilient plants.

Please clarify the following points from point of view of plant physiology:

Why was the seedling stage (younger stage) chosen for the experiment?

Although it is stated that the seedling stage was used for the microarray experiment data sets, more detailed information could be added to declare the age of the seedlings that were used. Additionally, it would be helpful to explain how the two different plants were harmonized at the seedling stage before carrying out the experiment. Since two different plants are being compared, it can be difficult to determine what stage of the seedling stage should be taken for next-generation sequencing or microarray experiments.

Can you please provide more detail on why you suggest that rice is more engaged in metabolism? What do you mean by this expression?

Can you explain why you decided on a two-fold cutoff for analyzing the up/down regulation of target genes?

For each plant, did you use four replicates?

6. PLOS authors have the option to publish the peer review history of their article (what does this mean?). If published, this will include your full peer review and any attached files.

Reviewer #1: No

Reviewer #2: No

Reviewer #3: **Yes: **Haniyeh Bidadi

---

## [Author Response · Author response to Decision Letter 0]

28 Apr 2023

Dear Editor-in-Chief to PLOS ONE,

I would like to express our deep appreciation to you and reviewers for providing us with the insight and direction needed to complete our submitted manuscript under the title of “Cold-responsive transcription factors in Arabidopsis and rice: A regulatory network analysis using array data and gene co-expression network”. Please kindly find the revised version of our manuscript according to the invaluable comments of reviewers. We have carefully taken reviewers’ comments and questions into the consideration and tried our best to fully address them both in the revised manuscript, as reflected here in our response to reviewers (following) and within our manuscript. We hope now the revised manuscript satisfies the reviewers, meets the standards of your respectful journal, and can be accepted for publication. 

Corresponding author

Dr. Naser Farrokhi

Associate Professor in Plant Molecular Biology,

Head of Department of Cell & Molecular Biology,

Faculty of Life Sciences & Biotechnology,

Shahid Beheshti University,

Tehran, Iran

+98 (21) 29905941

 

Reviewer #1: This manuscript deals with the exploration of cold-responsive transcription factors in Arabidopsis and rice. These results were preliminarily inferred ones which needs to be further analyzed for deriving sound concepts. The text seems mainly descriptive some TF and miRNA genes while the major scientific question to be answered is to be clarified. The authors should mention clearly the major scientific findings or progresses in comparison to the previously reported results.

• The manuscript comes from a bioinformatics point of view. However, it was revised to bring the findings more into light for further clarification.

Abstract

*Line 11-14 in Page 2. A clear conclusion is missed in the abstract. Please make definitive conclusions what authors really achieved from their results.

• A clear conclusion is added to the abstract:

According to the results, identified common TFs in rice and Arabidopsis have different regulatory networks at transcriptional and post-transcriptional levels. The regulatory mechanism of each identified TF in Arabidopsis and rice at transcriptional level were different in terms of interacting partners, co-expressed genes and as a result in downstream regulatory networks and metabolic pathways, so that identified cold-responsive TFs in rice seemed to be more engaged in energy metabolism esp. photosynthesis. Whereas, identified cold-responsive TFs in Arabidopsis were involved in signal transduction. At post-transcriptional level, miR5075 showed to target many TFs in rice. In comparison, the predictions showed that TFs are being targeted by diverse groups of miRNAs in Arabidopsis. Novel TFs, miRNAs and co-expressed genes were introduced as cold-responsive markers that can be harnessed in future studies and development of crop tolerant varieties.

Introduction

*Line 4 in Page 5. The common and specific mechanisms of cold tolerance in rice and Arabidopsis should be elaborated.

• The common and specific mechanisms of cold tolerance in rice and Arabidopsis is added to the manuscript:

In the past two decades, the molecular mechanisms of cold stress responses have been extensively studied in rice and Arabidopsis. A well-known transcriptional regulatory pathway involved in plant cold adaptation is the CBF/DREB1 cold signalling pathway mediated by CBF transcription factors (Chinnusamy and Zhu, 2007). Cold-activated CBF transcription factors (CBF1 to CBF3) are inducing cold response genes, which recognize and bind to the C-repeat/dehydration responsive element (CRT/DRE) motif in the promoters of many cold-responsive (COR) genes such as COR15A , COR15B and RD29a [8]. These genes are strongly upregulated in a CBF-dependent manner and which enhance the freezing resistance by stabilizing the chloroplast membranes when constitutively (over) expressed (Thalhammer et al., 2014). Vyse et al. [7] reported four TFs, CBF2/DREB1C, CBF4/DREB1D, DDF2/DREB1E and DDF1/DREB1F to be uniquely and significantly induced throughout the entire cold response. Given the fact that only ~12% of the cold-regulated genes are regulated by CBFs (Park et al., 2018), one has to assume that also other transcription factors are of importance for plant cold acclimation. 

• Chinnusamy V, Zhu J, Zhu JK. Cold stress regulation of gene expression in plants. Trends in plant science. 2007 Oct 1;12(10):444-51.

• Thalhammer A, Bryant G, Sulpice R, Hincha DK. Disordered cold regulated15 proteins protect chloroplast membranes during freezing through binding and folding, but do not stabilize chloroplast enzymes in vivo. Plant physiology. 2014 Sep;166 (1):190-201.

• Park S, Gilmour SJ, Grumet R, Thomashow MF. CBF-dependent and CBF-independent regulatory pathways contribute to the differences in freezing tolerance and cold-regulated gene expression of two Arabidopsis ecotypes locally adapted to sites in Sweden and Italy. PLoS One. 2018 Dec 5;13(12):e0207723.

*Lines 11-13 in Page 5. Authors have to cite examples on the effect of miRNA in cold resistance. Authors should cite some literature about the interaction mechanism of miRNA and mRNA in cold response.

• Some examples on the effect of miRNA in cold resistance and some literature the interaction mechanism of miRNA and mRNA in cold response are added to the manuscript:

There are some examples for miRNA and their predicted targets involved in regulation of rice and Arabidopsis growth and development under low-temperature stress. Overexpression of miR1320 resulted in increased cold tolerance in rice. AP2/ERF TF OsERF096, as a target of miR1320, co-regulate cold tolerance by repressing the JA-mediated cold signaling pathway (Sun et al., 2022). Similarly, overexpression of Osa-miR156, Osa-miR319, and Osa-miR528 also can improve cold resistance in rice (Huo et al.,2022). In addition, miR319 positively regulates cold tolerance by targeting OsPCF6 and OsTCP21 in rice, and the downregulation of these two transcription factors resulted in enhanced tolerance to cold stress (Wang et al., 2014). Recent research in Arabidopsis roots reported that Aux/IAA14 regulates miRNA-mediated cold stress responding mechanism. Based on next-generation sequencing, 180 known and 71 novel cold-responsive miRNAs were revealed. Furthermore, comparative analysis of miRNA expression shows notable difference of 13 known and 7 novel miRNAs in slr1 (mutation in Aux/IAA14) and wild types. Interestingly, compared with wild type, miR169 was downregulated in slr1 after 12-h cold treatment at 4◦C, particularly in the miR169a, miR169d, and miR169h (Zhang et al., 2022). The studies based on stress-response miRNAs can provide important understanding into plant stress resistance breeding and gene expression, a powerful approach to unravel new insight into adaptive mechanism in plants (Zhang et al., 2022).

• Sun M., Shen Y., Chen Y., Wang Y., Cai X., Yang J., et al.. (2022). Osa-miR1320 targets the ERF transcription factor OsERF096 to regulate cold tolerance via JA-mediated signaling. Plant Physiol. 189, 2500–2516. 10.1093/plphys/kiac208 

• Huo C., Zhang B., Wang R. (2022). Research progress on plant noncoding RNAs in response to low-temperature stress. Plant Signal. Behav. 17, 2004035. 10.1080/15592324.2021.2004035.

• Wang S. T., Sun X. L., Hoshino Y., Yu Y., Jia B., Sun Z. W., et al.. (2014). MicroRNA319 positively regulates cold tolerance by targeting OsPCF6 and OsTCP21 in rice (Oryza sativa L.). PLoS ONE 9, e91357. 10.1371/journal.pone.0091357 

• Zhang F, Yang J, Zhang N, Wu J, Si H. Roles of microRNAs in abiotic stress response and characteristics regulation of plant. Front Plant Sci. 2022 Aug 26;13:919243. doi: 10.3389/fpls.2022.919243.

Methods

*Line 5-7 in Page 6. What are the criteria for selecting a DET? Was it detected simultaneously by the 8 GO datasets or by any one of them?

• GEO2R was used to profile individual dataset lists of transcripts with significant increase and decrease in abundance compared to the untreated control condition. They were individually detected and then the common up- and down-regulated TFs were separated using Venn diagram.

• Differentially expressed transcripts (DETs) of TFs were defined with greater than two-fold change compared to the controls. We chose a two-fold cutoff according to the reasons bellow:

A two-fold change in gene expression is often considered biologically significant as it represents a substantial change in the level of gene expression. It is generally believed that changes in gene expression below this threshold may not have a significant functional impact on cellular processes. Therefore, using a two-fold cutoff helps to filter out relatively small changes in gene expression that may not be biologically relevant, and focuses on genes that exhibit more substantial changes in expression.

In addition, a two-fold cutoff reduces the impact of random variability and experimental noise statistically. Setting a fold-change cutoff minimizes the inclusion of genes that may show small changes in expression due to experimental variability or technical noise, which can be common in high-throughput gene expression data. By using a two-fold cutoff, it is more likely to capture genes that exhibit consistent and significant changes in expression across replicates, increasing the confidence in the results.

Moreover, using a two-fold cutoff for differential gene expression analysis enhances the reproducibility of results across different experiments or laboratories. It allows for consistent identification of significantly upregulated or downregulated genes, regardless of variations in experimental conditions, platforms, or data analysis methods. This helps to ensure that the findings are robust and reliable, and can be validated in independent experiments. It also helps to reduce the number of genes that need to be further analyzed or validated. Setting a higher fold-change cutoff, such as four-fold or higher, may result in a very small number of genes passing the threshold, which may not be practical for downstream analyses or functional validation. Therefore, a two-fold cutoff strikes a balance between sensitivity and specificity, allowing for a manageable number of genes for further investigation.

Therefore, we chose a two-fold cutoff for analyzing the up/down regulation of target genes.

• Ritchie ME, Phipson B, Wu DI, Hu Y, Law CW, Shi W, Smyth GK. limma powers differential expression analyses for RNA-sequencing and microarray studies. Nucleic acids research. 2015 Apr 20;43(7):e47.

• Khatri P, Sirota M, Butte AJ. Ten years of pathway analysis: current approaches and outstanding challenges. PLoS computational biology. 2012 Feb 23;8(2):e1002375.

Results

*Line 13 in Page 10. As shown in Figure 6, what are the differences between the GO terms for down- and up-regulation TFs? Most of them do not show a clear distinction between the two, so could down- and up-regulated TFs be combined to perform GO analysis?

• The “Figure 6” do not indicate the Gene ontology results of identified TFs in rice and Arabidopsis, but the Gene ontology results of co-expressed genes of identified up – and down- regulated TFs in rice and Arabidopsis in response to cold stress, which include different genes from different categories of cellular components, molecular function and biological process. The co-expressed genes of up-and down- regulated TFs are different from each other and each are obtained separately from databases. Co-expressed genes for each TF were retrieved from AttedII [35] for Arabidopsis, and RiceFREND [36] for rice according to MR values greater than 50 (Tables will be provided upon request). Therefore, they could not be combined to perform GO analysis and in order to have the possibility to a better comparison, all the results are presented in one figure.

*Line 16 in Page 10, Lines 1-2 in Page 11.The pathway results should be provided in the Supplementary Table.

• The metabolic pathway results are presented in Table S13 “Supplementary table 13” for co-expressed genes of Arabidopsis TFs and in Table S14 “Supplementary table 14” for co-expressed genes of rice. Both are added to the manuscript.

*Line 9 in Page 12. How does Figure 7 differ from Figure 2 in terms of the construction method? Authors need to provide additional details.

• It was just a typo. The correct reference to the figure is “Figure 7”. It was revised and added to the manuscript:

In order to identify the significant hub proteins having interaction with identified TFs, a PPI network was constructed. The most significant proteins with high rank interaction with TFs were UNE12 (AT4G02590) and NPR1 (AT4G19660), bHLH (AT1G03040), TGA-bzip (AT5G06950), NF-YB3 (AT4G14540), and bHLH105/ILR3 (AT5G54680) (Figure 7).

• Figure 2 and 7, both indicate the protein-protein interaction, so the construction method are the same. But they differ in terms of input materials. Figure 2 indicates the protein-protein interaction of up- and down-regulated TFs in Arabidopsis. But Figure 7, indicates the protein-protein interaction network of identified TFs with other proteins such as NPR1. These TF regulatory interactions were retrieved from CORNET using both experimental and predicted data of IntACt, TAIR and AtPID regulatory interactions (Figure 7, Table S5).

• 

*Line 6-8 in Page 14. The results should be provided to illustrate this conclusion.

• The results were added to the manuscript:

Six TFs in both plants were predicted to be targeted by only one specific miRNA in each plant including ANT (ath-miR5020c, osa-miR6255), ERF5 (ath-miR414, osa-miR437), bHLH35 (ath-miR1886.1, osa-miR414), bHLH137 (ath-miR5023, osa-miR5515), NFYA10 (ath-miR836, osa-miR2873a), and GATA23 (ath-miR5020b, osa-miR168b) (Table 2).

Discussion

*Line 1-2 in Page 15. Authors have mentioned that "In this study, PPI network analysis showed that some TFs, i.e., NFYA4, NFYA10, and NFYC2, could be considered as the hub genes in both plants. Combined with the previous studies, discuss the results with respect to the roles of these hub genes in cold stress.

• The results with respect to the roles of the hub genes in cold stress were discussed and added to the manuscript:

In this study, PPI network analysis showed that some TFs, i.e., NFYA4, NFYA10, and NFYC2, could be considered as the hub genes in both plants (Figure 2 & 3). Nuclear Factor-Y (NF-Y), composed of three subunits NF-YA, NF-YB and NF-YC, regulates the expression of target genes by directly binding the promoter CCAAT box or by physical interaction and mediating the binding of a transcriptional activator or inhibitor. NF-Y plays an important role at various stages of plant growth and development, especially in response to stress, which attracted many researchers to explore (Zhang et al., 2023).

Here, according to the analysis of TF co-expressed genes, it seems that NFY TFs in rice has regulatory effect on energy metabolism and biosynthesis of secondary metabolites (Figure 3, Table S14), whereas in Arabidopsis, NFY TFs affect biosynthesis of secondary metabolites and carbohydrates metabolism with fewer genes. This might be due to the smaller genome size of Arabidopsis (Figure 2, Table S13). Previous studies revealed that NF-Y members were involved in the stress response. In rice, OsNF-YA1 was down-regulated under both drought and cold stress and OsNF-YA5 was down-regulated in response to cold treatment (Yang et al., 2017). Arabidopsis NF-Y has an important role in the responses to abiotic stresses [52]. Kreps et al., (2002) identified Arabidopsis NF-YB2 through microarray analysis to be up-regulated by NaCl, mannitol, or cold (4℃) treatment. Hackenberg et al., (2012) reported the transcript level of AtNF-YC2 was highly induced by light, oxidative, heat, cold, and drought stress, while NF-YC4 was also induced by cold. NF-YB2 expression in Arabidopsis thaliana seedlings (16-day-old) was downregulated during early (0.5, 1 and 3 h) cold stress response while upregulated at the later stages (6, 12 and 24 h). Similar switching behaviour was displayed by AtNF-YB4 and AtNF-YB8, revealing these genes to play a putative role in late stages of plant adaptation to cold (Bhattacharjee et al., 2023). 

• Zhang, H.; Liu, S.; Ren, T.; Niu, M.; Liu, X.; Liu, C.;Wang, H.; Yin, W.; Xia, X. Crucial Abiotic Stress Regulatory Network of NF-Y Transcription Factor in Plants. Int. J. Mol. Sci. 2023, 24, 4426. https://doi.org/10.3390/ijms24054426

• Kreps JA, Wu YJ, Chang HS, Zhu T, Wang X, et al. (2002) Transcriptome changes for Arabidopsis in response to salt, osmotic, and cold stress. Plant Physiol 130: 2129–2141.

• Hackenberg D, Keetman U, Grimm B (2012) Homologous NF-YC2 subunit from Arabidopsis and tobacco is activated by photooxidative stress and induces flowering. Int J Mol Sci 13: 3458–3477.

• Bhattacharjee B and Hallan V (2023) NF-YB family transcription factors in Arabidopsis: Structure, phylogeny, and expression analysis in biotic and abiotic stresses. Front. Microbiol. 13:1067427. doi: 10.3389/fmicb.2022.1067427

• Wenjie Yang, Zhanhua Lu, Yufei Xiong, Jialing Yao, Genome-wide identification and co-expression network analysis of the OsNF-Y gene family in rice, The Crop Journal, Volume 5, Issue 1, 2017, Pages 21-31, ISSN 2214-5141, https://doi.org/10.1016/j.cj.2016.06.014.

*Line 14 in Page 15. A comparison of metabolic pathways (and hub genes) between rice and Arabidopsis should be elucidated.

• The comparison of the co-expressed gene network and their hubs in the rice and Arabidopsis indicated participation in different metabolic pathways. As mentioned in this section, the hub genes and other co-expressed genes of identified TFs in rice in this study are involved in photosynthesis and energy metabolism, lipid metabolism, biosynthesis of secondary metabolites, folding, sorting and degradation and transcription, terpenoids and polyketides metabolism, and circadian rhythm. Whereas, the most significant hubs in the Arabidopsis co-expressed gene network were transcription factors such as WRKY40, WRKY33, ZAT10, and ZAT12. In continue, we explained each pathway for co-expressed genes of both up and down regulated TFs in both plants in details. 

*Line 8-14 in Page 16. Based on your findings, please discuss the role of detected kinase in cold response rather than describing the conclusions of previous studies.

• The role of detected kinases in cold response is added to the manuscript as below:

Different Protein kinases were detected in cold response in rice and Arabidopsis such as MAP (Mitogen-Activated Protein) Kinase and LRR receptor-like serine/threonine-protein kinase. The role of these protein kinases in cold response has been studied in rice (Oryza sativa) and Arabidopsis thaliana. MAP Kinases are a class of protein kinases that play important roles in signal transduction pathways, including those involved in plant responses to various stresses, including cold stress (Xiong and Yang, 2003).

Studies have shown that MAP Kinases play a crucial role in the cold response pathway. In rice, activation of MAP Kinases plants upon exposure to cold stress leads to the phosphorylation of downstream target proteins, which turn trigger various cellular and molecular responses, such as changes in gene expression, accumulation of osmoprotectants, and modulation of ion transporters to cope with cold stress (Xiong and Yang, 2003) MAP Kinases in rice have been found to interact with other cold-responsive proteins and transcription factors, forming a complex regulatory network that modulates the plant's response to cold stress (Zhang et al., 2017).In cold stress, MAP Kinases in Arabidopsis are activated and regulate downstream targets, leading to changes in gene expression and various physiological responses, such as alterations in lipid metabolism, accumulation of osmoprotectants, and induction of antioxidant defense mechanisms (Liu and Zhang, 2004). 

The other protein kinases, LRR receptor-like kinases, are a type of receptor proteins that play a key role in many abiotic stress and physiological processes such as regulating gene expression responses and sensing external signals at the cellular environment level (Liao et al., 2017). For example, in rice, the expression of OsLRR2 in the leaves at the seeding, booting and flowering stage were markedly up-regulated after cold and drought treatment (Liao et al., 2017). The COLD1 (COLD REGULATED 1), a LRR receptor-like kinase in Arabidopsis, has been shown to play a crucial role in cold perception and signaling. COLD1 regulates the expression of C-repeat binding factors (CBFs), which are key transcription factors involved in cold response, leading to changes in gene expression and cold tolerance in Arabidopsis (Ma et al., 2015). 

• Yongrong Liao, Changqiong Hu, Xuewei Zhang, Xufeng Cao, Zhengjun Xu, Xiaoling Gao, Lihua Li, Jianqing Zhu & Rongjun Chen (2017) Isolation of a novel leucinerich repeat receptor-like kinase (OsLRR2) gene from rice and analysis of its relation to abiotic stress responses, Biotechnology & Biotechnological Equipment, 31:1, 51-57, DOI: 10.1080/13102818.2016.1242377

• Xiong L, Yang Y. (2003). Disease resistance and abiotic stress tolerance in rice are inversely modulated by an abscisic acid-inducible mitogen-activated protein kinase. Plant Cell, 15(3), 745-759 

• Zeyong Zhang, Junhua Li, Fei Li, Huanhuan Liu, Wensi Yang, Kang Chong, Yunyuan Xu, OsMAPK3 Phosphorylates OsbHLH002/OsICE1 and Inhibits Its Ubiquitination to Activate OsTPP1 and Enhances Rice Chilling Tolerance,Developmental Cell,Volume 43, Issue 6,2017,Pages 731-743.e5,ISSN 1534-5807,https://doi.org/10.1016/j.devcel.2017.11.016

• Markus Teige, Elisabeth Scheikl, Thomas Eulgem, Róbert Dóczi, Kazuya Ichimura, Kazuo Shinozaki, Jeffery L. Dangl, Heribert Hirt. The MKK2 Pathway Mediates Cold and Salt Stress Signaling in Arabidopsis. Molecular Cell,Volume 15, Issue 1,2004,Pages 141-152,ISSN 1097 2765,https://doi.org/10.1016/j.molcel.2004.06.023

• Liu Y, Zhang S. Phosphorylation of 1-aminocyclopropane-1-carboxylic acid synthase by MPK6, a stress-responsive mitogen-activated protein kinase, induces ethylene biosynthesis in Arabidopsis. The Plant Cell. 2004 Dec;16(12):3386-99.

• Zhao C, Nie H, Shen Q, et al. (2017). Phosphorylation of ICE1 Protein by JUN N-terminal kinase 1 Improves Freezing Tolerance in Arabidopsis. J Biol Chem, 292(11), 4559-4571

• Ma Y, Szostkiewicz I, Korte A, Moes D, Yang Y, Christmann A, Grill E. Regulators of PP2C phosphatase activity function as abscisic acid sensors. Science. 2009 May 22;324(5930):1064-8.

*Line 1-4 in Page 17. Authors described too many results in the discussion section with few references and little analysis. Authors should discuss the results with the appropriate literature.

• The results were discussed more and added to the manuscript as bellow:

The investigation of proteins interacting with TFs is of great importance. It has been shown that TFs interact with other TFs to form functional protein complexes [65]. Also, kinases may interact with TFs to act as a molecular switch to toggle their activities via phosphorylation [66] and many TFs form functional complexes like some NAC TFs and MADS TFs which form homo- or hetero-dimeric or tetrameric complexes (Heazlewood et al. 2007). Combinatorial interactions between transcription factors are important for the regulation of downstream genes (Kato et al., 2004). For example, in this study, there is the indirect interaction between bHLH105/ILR3 and bHLH59 with KNAT7 (Homeobox protein knotted-1-like 7), indicating potential cross-family interactions between different types of TFs. The KNAT7 is a Class II KNOTTED1-like homeobox (KNOX2) transcription factor gene that, in inter fascicular fibres, acts as a negative regulator of secondary cell wall biosynthesis (Wang et al., 2020). The cell wall is clearly affected by many abiotic stress conditions. A common plant response is the production of ROS and an increase in the activity of peroxidases, XTH (The xyloglucan endotransglucosylases/hydrolases) and expansins (Tenhaken,2015). KNAT7 forms a functional complex with OFP proteins to regulate aspects of secondary cell wall formation and OFP6 confers resistance to drought and cold stress in plants like rice (Ma et al., 2017). Li et al (2011) propose that KNAT7 forms a functional complex with OFP proteins to regulate aspects of secondary cell wall formation. They reported that AtOFP1 and AtOFP4 are components of a putative multi-protein transcription regulatory complex containing BLH6 and KNAT7 to regulate the formation of the secondary cell wall. So, our data revealed that TF interactions can also occur between different types of TF families, suggesting potential cross-talk and crosstalk regulatory mechanisms in transcriptional regulation.

The other interesting example of such cross-family interactions is the indirect interaction between NPR1 (Nonexpressor of Pathogenesis-Related Genes 1), which is a transcription co-activator involved in plant defense responses [67], and several TFs including Four members of ERF family (DREB 1A, DREB 1B/CBF1, ERF 4, ERF 113), three members of bHLH family (bHLH148, BIM2, UNE10) and MYB59 (Table S5). NPR1 (Nonexpressor of PR genes) is an essential regulator of plant systemic acquired resistance (SAR), which confers immunity to spectrum of pathogens (Mou et al., 2003). Singh et al. (2014) reported that 7 days of repetitive cold stress (1.5 hr at 4°C day−1) activated the pattern‐triggered immunity in Arabidopsis plants. Similarly, Kim et al. (2017) detected increased disease resistance in 3 weeks of cold stressed Arabidopsis plants, indicating NPR1 is partially required for cold activation of disease resistance, and there exists an NPR1‐independent SA pathway in cold activated immunity, similar to previous evidence showing that there is an NPR1‐independent SA pathway in plant defence response. It is suggested that the short‐term cold stress can act as a priming stimulus to prime defence response of Arabidopsis to bacterial pathogens (Wu et al., 2019). Taking these notions into account, it could be concluded that there is a crosstalk between cold stress and immunity. The results of our study indicate that while TFs (transcription factors) generally tend to interact with other TFs from their own family, it does not mean that interactions with other families should be ignored.

• Heazlewood JL, Verboom RE, Tonti-Filippini J, Small I, Millar AH. SUBA: the Arabidopsis subcellular database. Nucleic Acids Res. 2007;35:D213–D21. 

• Kato M, Hata N, Banerjee N, Futcher B, Zhang MQ. Identifying combinatorial regulation of transcription factors and binding motifs. Genome Biol. 2004;5(8):R56. doi: 10.1186/gb-2004-5-8-r56. Epub

• Wang S, Yamaguchi M, Grienenberger E, Martone PT, Samuels AL, Mansfield SD. The Class II KNOX genes KNAT3 and KNAT7 work cooperatively to influence deposition of secondary cell walls that provide mechanical support to Arabidopsis stems. Plant J. 2020 Jan;101(2):293-309. doi: 10.1111/tpj.14541.

• Tenhaken R. Cell wall remodeling under abiotic stress. Front Plant Sci. 2015 Jan 7;5:771. doi: 10.3389/fpls.2014.00771

• Yamei Ma, Chao Yang, Yong He, Zhihong Tian, Jianxiong Li, Rice OVATE family protein 6 regulates plant development and confers resistance to drought and cold stresses, Journal of Experimental Botany, Volume 68, Issue 17, 13 October 2017, Pages 4885–4898, https://doi.org/10.1093/jxb/erx309

• Li E, Wang S, Liu Y, Chen JG, Douglas CJ. OVATE FAMILY PROTEIN4 (OFP4) interaction with KNAT7 regulates secondary cell wall formation in Arabidopsis thaliana. Plant J. 2011 Jul;67(2):328-41. doi: 10.1111/j.1365-313X.2011.04595.x.

• Singh, P., Yekondi, S., Chen, P. W., Tsai, C. H., Yu, C. W., Wu, K., & Zimmerli, L. (2014). Environmental history modulates Arabidopsis pattern-triggered immunity in a HISTONE ACETYLTRANSFERASE1-dependent manner. The Plant Cell, 26(6), 2676– 2688. https://doi.org/10.1105/tpc.114.123356

• Kim YS, An C, Park S, Gilmour SJ, Wang L, Renna L, Brandizzi F, Grumet R, Thomashow MF. CAMTA-mediated regulation of salicylic acid immunity pathway genes in Arabidopsis exposed to low temperature and pathogen infection. The Plant Cell. 2017 Oct;29(10):2465-77.

• Wu Z, Han S, Zhou H, Tuang ZK, Wang Y, Jin Y, Shi H, Yang W. Cold stress activates disease resistance in Arabidopsis thaliana through a salicylic acid dependent pathway. Plant, cell & environment. 2019 Sep;42(9):2645-63.

• Mou Z, Fan W, Dong X. Inducers of plant systemic acquired resistance regulate NPR1 function through redox changes. Cell. 2003 Jun 27;113(7):935-44.

*Line 7-10 in Page 18. “TF-miRNA interactions seem to be different in Arabidopsis and rice in response to cold stress”. Discuss the results with respect to the role of TF-miRNA interactions in cold stress.

• The results were discussed with respect to the role of TF-miRNA interactions in cold stress and added to the manuscript:

The abiotic stress response network mediated by miRNA is one of the important mechanisms of plant response to various abiotic stresses. The miRNAs are implicated in abiotic stress response mechanisms with regard to oxidative stress and effects on DNA in different plant species (Pagano et al., 2021). Here, TF-miRNA interactions seem to be different in Arabidopsis and rice in terms of number of miRNA and mode of action in response to cold stress (Table 2). We found that the numbers of responsive miRNAs to cold stress in rice were greater than Arabidopsis. According to the results miR5075 targets most TFs in rice, while TFs in Arabidopsis are regulated by diverse sets of miRNA (Table 2). In addition, translation halt was the preferred mode of action in the post-transcriptional regulation mechanism in both plants. 

Transcription factors (TFs) and microRNAs play an important role in regulating the activity of the genes at transcriptional and post-transcriptional levels, respectively, involving a complex series of events (Li and Zhang, 2016; O’Brien et al., 2018). Under cold stress, variations in miRNAs expression (either up- or down-regulation) modify the transcript abundance of their target genes (Jeong and Green 2013; Zhang et al. 2014b; Nigam et al. 2015). For example, overexpression of rice miRNA156 was resulted in an increase in cell viability and growth rate under cold stress in rice and other plants through targeting OsSPL3 and other TFs [72]. According to their targets, miRNAs respond to low temperature stress through three tactics: the first is respond to abiotic stress directly; the second is indirectly responding to external stimuli by regulating transcription factors that relate to stress responses; and the third is that miRNAs can respond to multiple stresses and their target genes could code certain hydrolases or oxidoreductases (Yang et al.,2017). 

In this study, 192 new miRNA targeting up- and down-regulated TFs were identified in rice. Some of the novel miRNAs in relation to cold stress were miR5075, miR2927, miR159a.2 and miR1846 (Table 2). Our findings were also in accordance with earlier studies. For instance, miR319 (reported by [73]) targets ERF38, bHLH79, MYB5, and TCP1. miR398b [74] targets ERF74 and miR528 [75] targets ERF73, DREB1B, and GATA22 (Table 2). Tang et al. (2019) demonstrated that the overexpression of rice miRNA528 increased cell viability, growth rate, antioxidants content, ascorbate peroxidase (APOX) activity, and superoxide dismutase (SOD) activity under low-temperature stress in Arabidopsis and rice [75]. Their results suggested that OsmiR528 increases low-temperature tolerance by modulating the expression of the corresponding TFs.

miRNAs regulate at post-transcriptional level, particularly transcription factor combined directly with conservative cis-regulatory promoter seems to be more general. However, since most of these target genes are transcription factors, the mechanism of miRNA involved in plant stress response is more complex. Based on this, plant miRNAs have emerged as the promising targets for crop improvement, because they can control intricate agronomic traits, which give a positive regulation for better yield, quality, and stress tolerance (Zhang et al., 2017). Transcription factors as one of the target genes of miRNAs have multiple transcriptional activation functions according to their subunits, which have paramount importance in regulating plant progress and acclimation, and another target is gene encoding proteins or enzymes involved in plant metabolic regulation (Li, 2015; Wang et al., 2016; Samad et al., 2017). 

Similarly, in Arabidopsis, TF- miRNA interactions have been implicated in the regulation of cold stress responses. For example, over-expression of miR402 brings more tolerance to salinity, drought, and cold stress in A. thaliana (Kim et al., 2010). We found that the number of reported miRNAs for cold stress in Arabidopsis were greater than rice; some of which has already reported in response to cold stress. For instance, some of the previously reported miRNAs were miR156 [76], miR165, miR168, miR169, miR171, miR172, miR319, miR393, miR396, miR397 [77], miR402 [78], miR408 [79], miR157, miR159, miR164, miR166, miR394, miR398 [80], miR394a [81], miR397a [82], miR402 [83]. Identified miRNAs in Arabidopsis, miR157, miR171, miR393, and miR396 were in accordance with the literature which in this study target NFYB3, MYB5, ERF98, and ERF74, respectively. Also, miR156 targets bHLH116/ICE1 and HSFA3.

• Samad, A. F. A., Sajad, M., Nazaruddin, N., Fauzi, I. A., Murad, A. M. A., Zainal, Z., et al. (2017). MicroRNA and transcription factor: key players in plant regulatory network. Front. Plant Sci. 8, 565. doi: 10.3389/fpls.2017.00565

• Pagano, L., Rossi, R., Paesano, L., Marmiroli, N., and Marmiroli, M. (2021). miRNA regulation and stress adaptation in plants. Environ. Exp. Bot. 184, 104369. doi: 10.1016/j.envexpbot.2020.104369

• Yang, X., Liu, F., Zhang, Y., Wang, L., and Cheng, Y. F. (2017). Coldresponsive miRNAs and their target genes in the wild eggplant species Solanum aculeatissimum. BMC Genomics 18, 1000. doi: 10.1186/s12864-017-4341-y

• Zhang, H., Zhang, J., Yan, J., Gou, F., Mao, Y., Tang, G., et al. (2017). Short tandem target mimic rice lines uncover functions of miRNAs in regulating important agronomic traits. Proc. Natl. Acad. Sci. U.S.A. 114, 5277–5282. doi: 10.1073/pnas.1703752114

• Kim, J. Y., Kwak, K. J., Jung, H. J., Lee, H. J., and Kang, H. (2010). MicroRNA402 affects seed germination of arabidopsis thaliana under stress conditions via targeting DEMETER-LIKE Protein3 mRNA. Plant Cell Physiol. 51, 1079–1083. doi: 10.1093/pcp/pcq072

• Jeong DH, Green PJ (2013) The role of rice microRNAs in abiotic stress responses. J Plant Biol 56:187–197.

• Nigam D, Kumar S, Mishra DC, Rai A, Smita S, Saha A (2015) .Synergistic regulatory networks mediated by microRNAs and transcription factors under drought, heat and salt stresses in Oryza Sativa spp. Gene 555:127–139

• O’Brien, J., Hayder, H., Zayed, Y., and Peng, C. (2018). Overview of microRNA biogenesis, mechanisms of actions, and circulation. Front. Endocrinol. (Lausanne). 9:402. doi: 10.3389/fendo.2018.00402

• Li, C., and Zhang, B. (2016). MicroRNAs in control of plant development. J. Cell. Physiol. 231, 303–313. doi: 10.1002/jcp.25125

*Line 3 in Page 20. A summary should be derived from the above analysis in the perspective of regulatory networks. Besides, Authors have to add the future concept of the study.

• The summary is revised and added to the manuscript:

We compared common up-and down-regulated TFs in rice and Arabidopsis in response to cold stress to provide a detailed investigation of the pathways and candidate TFs. We tried to predict the potential target genes of cold-responsive TFs through co-expression network to uncover the regulatory networks involved in cold stress in Arabidopsis and rice. The construction of regulatory networks of TFs provides a comprehensive view of the molecular mechanisms underlying cold stress response. The results showed a significantly different regulatory mechanism of each TF in each plant in terms of co-expressed genes, interacting partners, downstream regulatory networks and pathways. In rice, the most significant hub genes were involved in photosynthesis. Whereas, in Arabidopsis the most significant hub genes were the TFs involved in signal transduction, suggesting that rice is more engaged in energy metabolism in contrast to Arabidopsis in response to cold. These finding have merits for further experimental analysis. Presented TFs, miRNAs and co-expressed genes in this study should be validated in terms of regulatory interactions between cold-responsive TFs and their target genes to confirm the functional relevance of the predicted regulatory networks. Knowledge about the regulatory networks of genes and proteins that define the cold-stress response is important in concepts of evolutionary biology among genera, helpful in defining subtle differences present within a species in response to varieties of stresses, and ultimately helpful towards the engineering of resilient plants before cold stress. Comparative transcriptional studies could also be used as a framework to investigate the regulatory networks of abiotic and abiotic stress responsive TFs in various plant species to contribute the advancement of plant stress biology research. 

Reviewer #2: The article entitled "Cold-responsive transcription factors in Arabidopsis and rice: A regulatory network analysis using array data and gene co-expression network" has chosen an important abiotic stress (cold) for investigation. Some comments are suggested to improve

the current version of this manuscript.

1. The description of materials and method needs to be revised. The descriptions of the data do not match the relevant tables completely. For example, the gene expression data of eight cold-treated microarray datasets (GEO) presented in table S1 are not only in the conditions of 4-5 °C, and 0 °C are also seen in these data. Therefore, different conditions may have different effects on the result of gene expression.

• “ 0-5 °C” is added to the manuscript.

Different cold stress treatments could lead to different results of gene expression. In general, low temperature stress includes 0–15°C and freezing stress (< 0°C), are defined as the synergy of low-temperature extremes beyond a plants optimal tolerance level (Xin, Z. & Browse, 2000; Penfield, 2008; Guo et al., 2018; Leuendorf et al., 2020). In this study the gene expression data of eight cold-treated microarray datasets were retrieved from GEO for A. thaliana and O. sativa in seedling stage treated for 24 h at 0 -5 °C. This temperature is considered as low temperature stress with the same effect on gene expression.

• Xin, Z. & Browse, J. 2000. Cold comfort farm: the acclimation of plants to freeying temperatures. Plant Cell Environ. 23, 893-902.

• Penfield, S. 2008.Temperature perception and signal transduction in plants. New Phytol. 179, 615-628.

• Guo X, Liu D, Chong K. Cold signaling in plants: Insights into mechanisms and regulation. J Integr Plant Biol. 2018 Sep;60(9):745-756. doi: 10.1111/jipb.12706. PMID: 30094919.

• Leuendorf, J.E., Frank, M. & Schmülling, T. Acclimation, priming and memory in the response of Arabidopsis thaliana seedlings to cold stress. Sci Rep 10, 689 (2020). 

2. It is better to provide correct and more complete explanations for the figures and tables of the manuscript.

• The titles of figures and tables are revised and added to the manuscript.

3. Which the authors claim in this report, new TFs, miRNAs and co-expressed genes have been introduced as cold-responsive markers, also the authors claim these cold-responsive markers can be used in future studies and the development of tolerant varieties. It would have been better to add a verification analysis or some kind of confirmation to this article. Because the number of introduced

genes, TFs, miRNAs is large and it is necessary to limit them in a way and to introduce cold-responsive markers. Especially, it is likely that what was introduced in this research is not specific to the conditions of cold stress and may have a different expression in other stresses, especially in abiotic stresses. Therefore, it is better to investigate and report the expression and behavior of introduced TFs, miRNAs and genes in other abiotic stresses such as drought and heat. If there are common in abiotic stresses, it is necessary to identify them, and according to the title of the article, cold-responsive transcription factors in Arabidopsis and rice should be specifically introduced.

• In this study, we tried to make an in silico comparison of Arabidopsis and rice TFs in response to cold stress. Although the confirmation and specification of the presented gene candidates in this study would absolutely improve the results and the quality of the article, but was not the purpose in this stage. But the results of the article provide a basis for further experimental analysis and the engineering of resilient plants, as we also mentioned in the conclusion.

Reviewer #3: This study provides a comparative analysis of the transcriptional regulatory response to cold stress in rice and Arabidopsis, with a focus on the identification of up- and down-regulated TFs and miRNAs. The results show differences in the number and diversity of TF

families in each plant, as well as differences in the regulatory mechanisms of each TF. Additionally, miRNAs in Arabidopsis were found to target TFs more specifically compared to rice. The study highlights the importance of understanding the regulatory networks involved in the response to cold stress in plants, and provides a basis for further experimental analysis and the engineering of resilient plants.

Please clarify the following points from point of view of plant physiology:

1. Why was the seedling stage (younger stage) chosen for the experiment?

• Low temperatures and frost compromise the plant survival and ultimately lead to growth retardation and yield loss. Many species of tropical or subtropical origin are injured or killed by nonfreezing low temperatures, and exhibit various symptoms of chilling injury such as chlorosis, necrosis, or growth retardation. In contrast, chilling-tolerant species are able to grow at such low temperatures. Rice (Oryza sativa L.), a major cereal crop, thrives in both tropical and temperate regions around the world. Rice (Oryza sativa L.) feeds more than half of the global population.

Cold stress tolerance is important throughout the life cycle of the rice plant, but especially in the early vegetative stages, i.e., at germination when the coleoptile elongates and as the young seedling develops. The damage caused by low temperatures at the seedling stage is mainly observed as leaf rolling, necrosis, chlorosis and stunting. When subjected to cold temperatures, seedlings demonstrate a wide range of genetic and physiological responses to protect their cell and plasma membranes, including activation of gene and protein expression, changes in membrane lipid composition, and accumulation of hydrophobic polypeptides.

Studying cold stress at the seedling stage allows researchers to investigate the physiological and molecular responses of plants to cold stress during a critical growth stage, which is more relevant to field conditions where young seedlings are exposed to cold stress during early growth. Hence, seedling stage was chosen to be analysed in this study.

• Das R, Mukherjee A, Basak S, Kundu P. Plant miRNA responses under temperature stress. Plant Gene. 2021 Dec 1;28:100317.

• Zhang J, Li J, Wang X, Chen J (2011) OVP1, a vacuolar H+-translocating inorganic pyrophosphatase (V-PPase), overexpression improved rice cold tolerance. Plant Physiol Biochem: 49: 33–38. 10.1016/j.plaphy.2010.09.014

• Andaya VC, Mackill DJ. Mapping of QTLs associated with cold tolerance during the vegetative stage in rice. J Exp Bot. 2003;54: 2579–2585. 10.1093/jxb/erg243

• Koseki M, Kitazawa N, Yonebayashi S, Maehara Y, Wang ZX, Minobe Y. Identification and fine mapping of a major quantitative trait locus originating from wild rice, controlling cold tolerance at the seedling stage. Mol Genet Genomics. 2010; 284: 45–54. 10.1007/s00438-010-0548-1

• Counce PA, Keisling TC, Mitchell AJ. A uniform, objectives, and adaptive system for expressing rice development. Crop Sci. 2000;40: 436–443

• Lou Q, Chen L, Sun Z, Xing Y, Li J, Xu X, et al. A major QTL associated with cold tolerance at seedling stage in rice (Oryza sativa L.). Euphytica. 2007; 158(1–2): 87–94.

• Cui S, Huang F, Wang J, Ma X, Cheng Y, Liu J. A proteomic analysis of cold stress responses in rice seedlings. Proteomics. 2005;5: 3162–3172.

2. Although it is stated that the seedling stage was used for the microarray experiment data sets, more detailed information could be added to declare the age of the seedlings that were used. Additionally, it would be helpful to explain how the two different plants were harmonized at the seedling stage before carrying out the experiment. Since two different plants are being compared, it can be difficult to determine what stage of the seedling stage should be taken for next-generation sequencing or microarray experiments.

• The age of the seedlings is added to the Table S1. All the seedlings were two- week old. 

• Yes. The seedling stage of rice and Arabidopsis as monocot and dicot plants are different. In this study, we chose the microarray data from totally different experiments, but with the same experimental condition of 24 hours 0-5 °C cold treatment in seedling stage of both model plants. But in the analysing stage of this study, we normalized the gene expression data of each microarray dataset.

3. Can you please provide more detail on why you suggest that rice is more engaged in metabolism? What do you mean by this expression?

• The most significant hubs in the rice co-expressed gene network were PSI-F, PSI-K, chloroplastic UPF0603, chloroplast photosystem I reaction center subunit, PSI-G and chloroplastic chlorophyll a-b binding protein (Figure 4). On the other hand, PPI and gene ontology of data showed that most of the co-expressed genes of cold-induced TFs in rice were involved in energy metabolism, lipid metabolism, biosynthesis of secondary metabolites, folding, sorting and degradation and transcription, terpenoids and polyketides metabolism, and circadian rhythm. However, the most significant hubs in the Arabidopsis co-expressed gene network were WRKY40, WRKY33, ZAT10, ZAT12 (Figure 5), which have been reported as TFs involved in cold stress. These results suggest that rice is more engaged in energy metabolism especially photosynthesis during cold stress.

4. Can you explain why you decided on a two-fold cutoff for analyzing the up/down regulation of target genes?

• A two-fold cutoff is commonly used in microarray and RNA-seq data analysis to determine the differential expression of target genes, where genes that show at least a two-fold change in expression are considered significantly upregulated or downregulated. We chose a two-fold cutoff according to the reasons bellow:

A two-fold change in gene expression is often considered biologically significant as it represents a substantial change in the level of gene expression. It is generally believed that changes in gene expression below this threshold may not have a significant functional impact on cellular processes. Therefore, using a two-fold cutoff helps to filter out relatively small changes in gene expression that may not be biologically relevant, and focuses on genes that exhibit more substantial changes in expression.

In addition, a two-fold cutoff reduce the impact of random variability and experimental noise statistically. Setting a fold-change cutoff minimizes the inclusion of genes that may show small changes in expression due to experimental variability or technical noise, which can be common in high-throughput gene expression data. By using a two-fold cutoff, it is more likely to capture genes that exhibit consistent and significant changes in expression across replicates, increasing the confidence in the results.

Moreover, using a two-fold cutoff for differential gene expression analysis enhances the reproducibility of results across different experiments or laboratories. It allows for consistent identification of significantly upregulated or downregulated genes, regardless of variations in experimental conditions, platforms, or data analysis methods. This helps to ensure that the findings are robust and reliable, and can be validated in independent experiments. It also helps to reduce the number of genes that need to be further analyzed or validated. Setting a higher fold-change cutoff, such as four-fold or higher, may result in a very small number of genes passing the threshold, which may not be practical for downstream analyses or functional validation. Therefore, a two-fold cutoff strikes a balance between sensitivity and specificity, allowing for a manageable number of genes for further investigation.

Therefore, we chose a two-fold cutoff for analyzing the up/down regulation of target genes.

• Ritchie ME, Phipson B, Wu DI, Hu Y, Law CW, Shi W, Smyth GK. limma powers differential expression analyses for RNA-sequencing and microarray studies. Nucleic acids research. 2015 Apr 20;43(7):e47.

• Khatri P, Sirota M, Butte AJ. Ten years of pathway analysis: current approaches and outstanding challenges. PLoS computational biology. 2012 Feb 23;8(2):e1002375.

5. For each plant, did you use four replicates?

• In this study, the total number of 16 microarray data set (8 for Rice and 8 for Arabidopsis) was applied and each microarray data was the result of different replications according to the experiment design of the original articles which are mentioned in the reference section.

---

## [Decision Letter · Decision Letter 1]

15 May 2023

Cold-responsive transcription factors in Arabidopsis and rice: A regulatory network analysis using array data and gene co-expression network

PONE-D-22-30309R1

Dear Dr. Farrokhi,

We’re pleased to inform you that your manuscript has been judged scientifically suitable for publication and will be formally accepted for publication once it meets all outstanding technical requirements.

Kind regards,

Keqiang Wu, Ph.D

Academic Editor

PLOS ONE

Additional Editor Comments (optional):

Reviewers' comments:

Reviewer's Responses to Questions

**Comments to the Author**

1. If the authors have adequately addressed your comments raised in a previous round of review and you feel that this manuscript is now acceptable for publication, you may indicate that here to bypass the “Comments to the Author” section, enter your conflict of interest statement in the “Confidential to Editor” section, and submit your "Accept" recommendation.

Reviewer #1: All comments have been addressed

Reviewer #3: All comments have been addressed

2. Is the manuscript technically sound, and do the data support the conclusions?

Reviewer #1: Yes

Reviewer #3: Yes

3. Has the statistical analysis been performed appropriately and rigorously? 

Reviewer #1: Yes

Reviewer #3: Yes

4. Have the authors made all data underlying the findings in their manuscript fully available?

Reviewer #1: Yes

Reviewer #3: Yes

5. Is the manuscript presented in an intelligible fashion and written in standard English?

Reviewer #1: Yes

Reviewer #3: Yes

6. Review Comments to the Author

Reviewer #1: (No Response)

Reviewer #3: (No Response)

7. PLOS authors have the option to publish the peer review history of their article (what does this mean?). If published, this will include your full peer review and any attached files.

Reviewer #1: No

Reviewer #3: **Yes: **Haniyeh Bidadi

---

## [Editor Report · Acceptance letter]

31 May 2023

PONE-D-22-30309R1 

Cold-responsive transcription factors in Arabidopsis and rice: A regulatory network analysis using array data and gene co-expression network 

Dear Dr. Farrokhi:

I'm pleased to inform you that your manuscript has been deemed suitable for publication in PLOS ONE. Congratulations! Your manuscript is now with our production department. 

Kind regards, 

on behalf of

Professor Keqiang Wu 

Academic Editor

PLOS ONE